# Suppression of trabecular meshwork phagocytosis by norepinephrine is associated with nocturnal increase in intraocular pressure in mice

Keisuke Ikegami [1]✉ & Satoru Masubuchi[1]

Intraocular pressure (IOP) is an important factor in glaucoma development, which involves aqueous humor (AH) dynamics, with inflow from the ciliary body and outflow through the trabecular meshwork (TM). IOP has a circadian rhythm entrained by sympathetic noradrenaline (NE) or adrenal glucocorticoids (GCs). Herein, we investigated the involvement of GC/NE in AH outflow. Pharmacological prevention of inflow/outflow in mice indicated a diurnal outflow increase, which was related to TM phagocytosis. NE showed a non-self-sustained inhibition in phagocytosis of immortalized human TM cells, but not GC. The pharmacological and reverse genetic approaches identified β1-adrenergic receptor (AR)-mediated exchange proteins directly activated by cyclic adenosine monophosphate (EPAC)-SHIP1 signal activation by ablation of phosphatidylinositol triphosphate, regulating phagocytic cup formation. Furthermore, we revealed the phagocytosis involvement in the β1-AR-EPAC-SHIP1-mediated nocturnal IOP rise in mice. These suggest that TM phagocytosis suppression by NE can regulate IOP rhythm through AH outflow. This discovery may aid glaucoma management.

---

[1] Department of Physiology, School of Medicine, Aichi Medical University, 1-1 Yazako-karimata, Nagakute, Aichi 480-1195, Japan.
✉email: ikegami.keisuke.910@mail.aichi-med-u.ac.jp

In most organisms, multiple physiological processes are controlled by circadian rhythms, which last approximately 24 h. The suprachiasmatic nucleus (SCN) in the anterior hypothalamus acts as a circadian pacemaker in mammals, which senses optical information from the retina, to manage daily body rhythms[1]. The SCN synchronizes most peripheral tissues and cells through various complex pathways, involving mainly the autonomic nervous system and endocrine signals[2]. Norepinephrine (noradrenaline, NE) released from the superior cervical ganglion (SCG), a part of the sympathetic nervous system, transmits circadian timing signals to the ciliary body of the eye to regulate pupil size, among other functions[3]. Furthermore, for most peripheral tissues, glucocorticoids (GCs) secreted from the adrenal glands via the hypothalamus-pituitary-adrenal axis-mediated SCN act as strong endocrine timing signals because GC receptors (GRs) are expressed in most peripheral cell types[4].

Glaucoma is a leading cause of blindness in elderly people; however, no effective cure exists. Abnormal intraocular pressure (IOP) inside the eye, for example, high IOP, contributes to glaucoma development and progression characterized by vision loss[5]. IOP is balanced by the aqueous humor (AH) and has a circadian rhythm. In humans, nocturnal IOP increases irrespective of posture[6]. IOP is also elevated at night in nocturnal or diurnal animals[7], which is controlled by the SCN in mice[8]. Nocturnal IOP is also elevated in patients with glaucoma[6,9], and the IOP rhythm undergoes phase shifts in patients with primary open-angle glaucoma (POAG) and normal-tension glaucoma (NTG)[9]. NTG was also reported to involve an abnormal IOP rhythm[10]. Aging desynchronizes the IOP rhythm in older healthy subjects[11]. Interestingly, IOP rhythm is disrupted in night-shift workers[12]. A recent study reported that a disrupted circadian IOP rhythm causes optic nerve damage and increases the risk of glaucoma[11]. Thus, regulation of IOP rhythm, especially during the night, is central to glaucoma management, and the circadian mechanism in AH dynamics is important for glaucoma therapy. We have previously demonstrated the dual pathway by which both NE and GCs transmit timing information to the eye to form the IOP rhythm in mice[13]. However, the molecular mechanisms of this axis remain unknown.

In AH dynamics, the non-pigmented epithelial cell (NPE) of the ciliary body participates in AH production[14]. In contrast, the trabecular outflow pathway is responsible for homeostatic regulation of IOP and is regulated by the coordinated generation of AH outflow resistance mediated by the constituent cells of the trabecular meshwork (TM) and Schlemm's canal (SC)[15,16], and partially, uveoscleral outflow is known to be involved in human IOP rhythm[17]. Most of this resistance is believed to be generated in the inner wall region comprising the juxtacanalicular tissue and the inner wall endothelium of the SC and its pores (small openings in the giant vacuoles)[18]. The mechanisms that regulate aqueous outflow resistance in normal and glaucomatous eyes remain unclear, and a few newly approved medications target this site of resistance. TM phagocytosis can decrease particulate material and debris from AH, attenuating outflow resistance, and contribute to IOP reduction[15]. Phagocytosis is thought to play an important role in the normal functioning of the outflow pathway by keeping the drainage channels free of debris. Conversely, actin remodeling, such as fragmentation or polymerization, also regulates AH outflow, and the actin cytoskeleton of TM cells is a therapeutic target in glaucoma patients. Small GTPase Rho-associated coiled-coil-containing protein kinase (ROCK) inhibitors lower IOP, not only by relaxation of the TM by disruption of actin stress fibers, but also by the activation of phagocytosis[19–21]. However, since all phagocytic processes are driven by a finely controlled rearrangement of the actin cytoskeleton[22,23], it has been difficult to verify the effects of these two factors separately.

Long-term dexamethasone (Dex) treatment is known to decrease TM phagocytosis and increase actin fiber stress in the human eye, as well as primary TM cells[24,25], leading to increased AH outflow resistance, and ultimately, glaucoma development. In addition, NE suppresses wound macrophage phagocytic efficiency through α- and β-adrenergic receptor (AR)-dependent pathways[26], and can promote actin polymerization in the retina[27]. However, the detailed effect of NE on TM phagocytosis has not yet been elucidated. Hence, this study aimed to uncover the effects of GC/NE on trabecular phagocytosis and its molecular mechanisms. We addressed the effect of AH outflow on diurnal IOP changes in mice and found the involvement of TM phagocytosis in the AH outflow. In addition, the effect of NE/GC exposure on phagocytic activity was monitored in real time using immortalized human TM cells. Pharmacological approaches and RNA interference have identified AR-related pathways involved in phagocytosis regulation. Furthermore, the role of the identified regulatory pathway in IOP rhythm regulation was assessed using pharmacological instillation in mice.

## Results

**AH outflow increased in the daytime in mice.** To address the effects of ciliary AH-production on nocturnal increase in IOP, we injected the $Na^+/K^+$ATPase antagonist, ouabain, into the anterior chamber of the mouse eye at zeitgeber time (ZT) 10 (Fig. 1a). Ouabain allowed a nocturnal IOP increase (Fig. 1b), but prevented IOP increase in individual data ($p < 0.01$) (Fig. 1c), indicating the effect of AH outflow. Since AH drainage suppression from the TM by microbeads intraocularly (i.o.) injection elevates IOP in rats[28], to identify the role of AH outflow on mouse IOP rhythm, microbeads were injected into the anterior chamber of the eye to mildly suppress the AH outflow (Fig. 1d), as previously reported[28]. After 2 weeks, IOP in bead-injected mice at ZT6 increased up to the nocturnal level (Fig. 1e), consistent with previous reports[28,29]. Day-night differences in individual IOP were arrested by bead injection (Fig. 1f), indicating a daytime decrease in IOP due to AH outflow. To confirm diurnal changes in AH drainage in the TM, we administered small fluorescent particles to the anterior chamber at ZT0 and ZT12, and subsequently observed the anterior eye extracted at ZT6 and ZT18, respectively, and measured the fluorescence intensity (Fig. 1g), where we assumed that the particles were taken into the SC or passed through the TM (Fig. 1h). The fluorescence intensity at ZT6 was significantly stronger than that at ZT18 (Fig. 1h, i), consistent with a previous rabbit AH outflow study[30]. Thus, it seems plausible that AH outflow through TM/SC may be involved in the daytime decrease in IOP in mice.

**TM phagocytosis is involved in diurnal IOP reduction.** Phagocytosis and remodeling of the actin cytoskeleton are related to AH outflow[15,19–21]. Dynasore, an inhibitor of dynamin GTPase activity, but not other small GTPases, has been widely studied in clathrin-mediated endocytosis and phagocytosis, also in HTMC[31] (Fig. 2a). Dynamin is thought to interact with actin filaments when the edges of the phagocytic cup close[32]. Latrunculin A and cytochalasin D, drugs that interfere with actin–myosin contraction, also lower IOP[33,34] and suppress TM phagocytosis[33–35] (Fig. 2a). To examine their involvement in daytime IOP reduction, we first administered mice dynasore, cytochalasin D, latrunculin A, and RKI1447 (ROCK inhibitor; phagocytosis enhancer and cytoskeleton disruptor [Fig. 2a]) at ZT4 and measured IOP at ZT9 (Fig. 2b). Instillation with dynasore significantly increased IOP, whereas latrunculin A decreased IOP (Fig. 2c). Individual data also showed that dynasore enhanced diurnal IOP (Fig. 2d), indicating that the daytime activation of

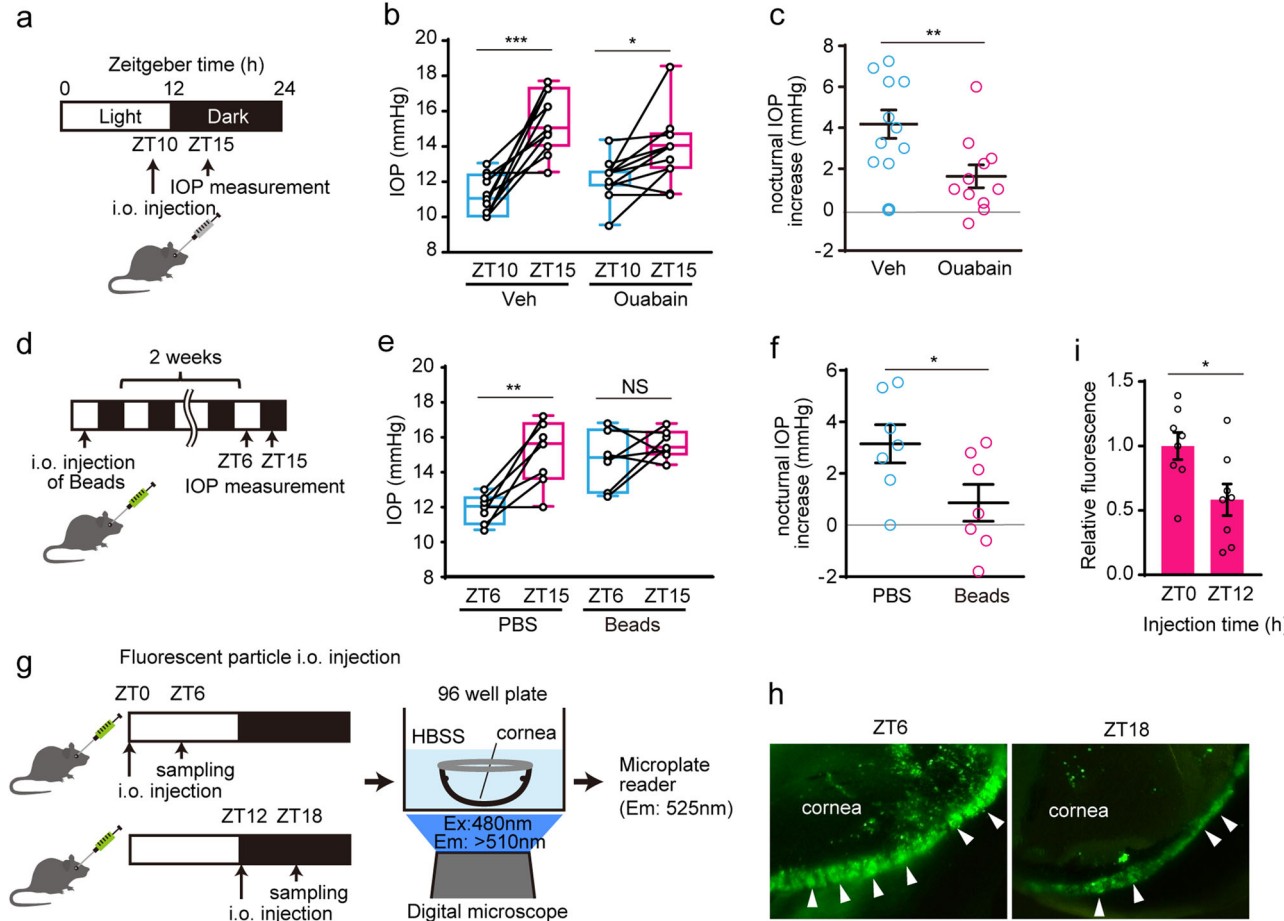

**Fig. 1 Day-time increase in aqueous humor outflow in mice. a–c** Effect of $Na^+/K^+ATPase$ antagonist, ouabain, treatment on nocturnal intraocular pressure (IOP) increase. **a** Ouabain was injected to anterior chamber of mouse eye at ZT10, and IOP were measured before injection and 5 h after injection (ZT15). **b** Ouabain allowed a nocturnal IOP increase (paired t-test, $*p < 0.05$, $***p < 0.001$). Data are presented as box-and-whisker plots with individual data ($n = 11$). Variability is shown in the box from the 25th to 75th percentiles, and min to max (whiskers). **c** Ouabain individually prevents a nocturnal IOP increase (t-test, $**p < 0.01$). Data are presented as scatter plots with mean ± SEM ($n = 11$). **d–f** Diurnal changes of IOP (ZT6 and ZT15) were measured 2 weeks after intraocular injection of beads. **e** IOP at ZT6 was increased up to nocturnal level (paired t-test, $**p < 0.01$). Data are presented as box-and-whisker plots with individual data ($n = 7$). **f** Day-night differences in individual IOP were arrested by bead injection. Data are presented as scatter dot plots with mean ± SEM ($n = 7$). **g–i** Fluorescent particles were injected into the anterior chamber of the eye at ZT0 or ZT12, and after 6 h; the fluorescence of extracted cornea was observed with a microscope (Ex: 480 nm, Em: >510 nm) and quantified by microplate reader (Em: 525 nm). **h** Diurnal change of the particles at the edge of cornea (speculating the TM) was detected (arrowhead). **i** Fluorescent intensity at ZT6 was significantly higher than that at ZT18 (t-test, $*p < 0.05$). Data are presented as bar graphs (mean ± SEM) with scatter dot plots ($n = 8$). ZT Zeitgeber time.

TM phagocytosis may suppress IOP, which can be canceled by depolymerization of actin fibers. In contrast, instillation with cytochalasin D, latrunculin A, and RKI1447 at ZT10 arrested the nocturnal IOP increase, especially latrunculin A significantly suppressed IOP, but not dynasore (Fig. 2e, f). At the individual level, these drugs, regardless of their effects on phagocytosis, also suppressed the nocturnal IOP rise (Fig. 2g), indicating the role of TM actin polymerization in nocturnal IOP rise. This IOP suppression limited to nighttime by ROCK inhibitor was similar to a previous report demonstrating that RhoA blocking by AAV prevents nocturnal IOP elevation in rats[36]. Taken together, these results suggest that AH outflow regulation by the cytoskeleton seems to be night-limited or time-independent, while TM phagocytosis may mediate the daytime increase in AH outflow.

**Driven-output stimuli of NE may suppress phagocytosis in the TM.** To investigate the effect of GC and NE on phagocytosis in the TM, we used pH-sensitive pHrodo particles to evaluate phagocytosis in real time in immortalized human TM cells

(iHTMCs) (Fig. 3a). We first validated these iHTMCs by analyzing gene expression (direct GC-induced clock gene PER1, trabecular meshwork-inducible GC response protein [MYOC][37], and GC-suppressed matrix metalloproteinase MMP3[38,39]), 6 h (early) and 48 h (late) after Dex stimulation. Early induction of PER1, late induction of MYOC, and late suppression of MMP3 were detected (Supplementary Fig. 1), indicating that this iHTMC has TM cell properties[40].

Next, the validity of this phagocytosis assay was confirmed using a control with Dynasore (Fig. 3b). After medium changes, including NE or Dex, phagocytosis was significantly suppressed by NE (Fig. 3c, d). When we performed an in vitro NE clearance assay, the half-life of medium NE (~18.6 h, $y = 335.9\ e^{-0.000621\ x}$, $r^2 = 0.979$) (Supplementary Fig. 2) was far greater than that of blood NE (a few minutes), suggesting that in vitro NE treatment continuously act on cells. Indeed, NE treatment dose-dependently prevented phagocytic activity in iHTMCs during 2 days of culture (Fig. 3d). Although Dex alone had no effect (Fig. 3e, f and Supplementary Fig. 3a), Dex with NE-suppressed phagocytosis in a dose-dependent manner (Supplementary Fig. 3b), consistent

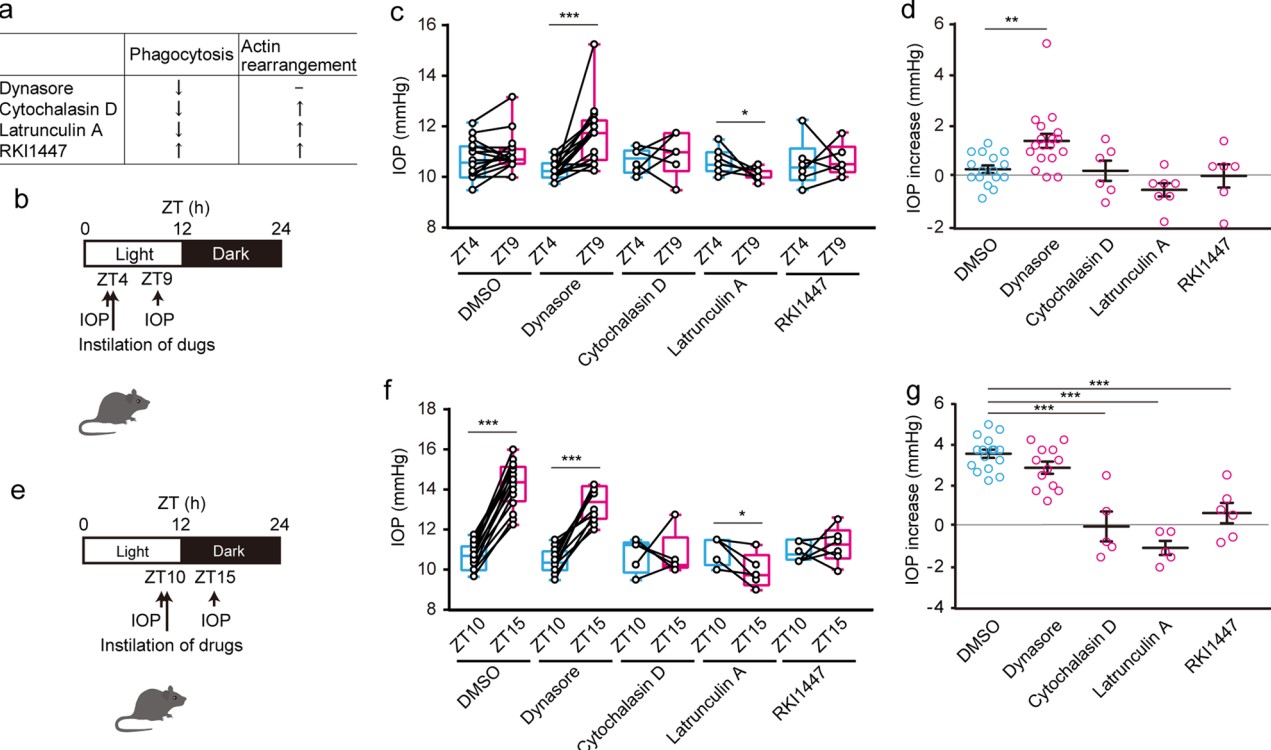

**Fig. 2 Involvement of trabecular meshwork (TM) phagocytosis on diurnal IOP reduction in mice. a–d** Effect of phagocytosis regulatory drugs (**a**) on diurnal IOP increase in mice. **b** Mice were administered an instillation of Dynamin-mediated phagocytosis inhibitor (Dynasore; 1 mM), inhibitors of actin polymerization and phagocytosis (cytochalasin D; 100 μM, latrunculin A; 100 μM), and cytoskeleton disruptor/phagocytosis activator (ROCK inhibitor RKI1447; 1 mM) at ZT4 and IOP were measured at ZT9. **c** Instillation with Dynasore significantly increased IOP, whereas latrunculin A decreased IOP (paired t-test, *$p < 0.05$, ***$p < 0.001$). Data are presented as box-and-whisker plots with individual data ($n = 6$–18). **d** Individual data also showed that Dynasore enhanced diurnal IOP (**$p < 0.01$, one-way ANOVA [$p < 0.001$], Dunnett's multiple comparison test). Data are presented as scatter plots with mean ± SEM ($n = 6$–18). **e–g** Effect of phagocytosis-regulatory drugs on nocturnal IOP increase in mice. Mice were administered an instillation of drugs at ZT10, and IOP was measured at ZT15. **f** Instillation of cytochalasin D, latrunculin A, and RKI1447 arrested the nocturnal IOP increase, especially latrunculin A significantly suppressed IOP (paired t-test, *$p < 0.05$, ***$p < 0.001$). Data are presented as box-and-whisker plots with individual data ($n = 5$–16). **g** All inhibitors except Dynasore prevented an increase in nocturnal IOP (***$p < 0.001$ vs. DMSO, one-way ANOVA [$p < 0.001$], Dunnett's multiple comparison test). Data are presented as scatter plots with mean ± SEM ($n = 5$–16).

with previous in situ studies[24,25]. Long-term Dex exposure did not alter cell viability (Supplementary Fig. 3c) or senescence (Supplementary Fig. 3d), indicating no effect of apoptosis or senescence on phagocytosis inhibition by Dex with NE. GC also binds to TM in humans[41], and its receptor localizes in mouse TM[13]. Thus, the interaction between GCs and NE can generate an appropriate AH drainage rhythm. Taken together, these results suggest the importance of NE in TM phagocytosis.

Circadian time signals can generate rhythmicity through self-sustainable autonomous rhythm by single stimulation or a driven-output system by stimulation of circadian factors such as NE and GCs once a day (Fig. 3g). Since NE from the SCG appears to show a nocturnal peak in rodents[42], the circadian rhythm of SCG-NE can regulate diurnal changes in TM phagocytosis. To investigate the short-term effect of NE and Dex on phagocytic circadian rhythm and activity, we exposed iHTMC to NE and Dex for 30 min, and subsequently observed the phagocytic activity in iHTMCs in real-time for 3 days (Fig. 3h). Although we could not observe the circadian phagocytosis rhythm in the normalized fluorescence signal by control, high NE pulse stimulation immediately began to suppress phagocytosis, reaching a minimum after 9 h and showing an inhibitory effect over 24 h, while Dex stimulation did not modulate phagocytosis rhythm (Fig. 3i). These findings suggest the possibility that diurnal changes in phagocytosis in TM may

occur as a result of a driven output, but are not self-sustainable after a single NE stimulation (Fig. 3g).

**β1-AR mainly attenuates phagocytosis in iHTMCs.** As NE suppresses wound macrophage phagocytic efficiency through α- and β-AR dependent pathways[26], we first confirmed the gene expression of nine AR subtypes in iHTMC. Strong expression of *ADRA2A*, *ADRA2B*, *ADRA2C*, *ADRB1*, and *ADRB2* was observed, while very weak expression of *ADRA1s* and *ADRB3* (Supplementary Fig. 4a) was observed, consistent with the gene expression patterns analyzed from previous single-cell RNA-seq data in human TM macrophages (Supplementary Fig. 4b)[43]. To identify the AR regulating SC phagocytosis, we subsequently treated several agonists of AR (L-adrenaline, β1-AR agonists [L-NE and dobutamine], β1β2-AR agonist [isoproterenol], selective α1-AR agonist [phenylephrine], direct-acting α2-AR agonist [clonidine], and β2-AR agonists [formoterol and salbutamol]) to iHTMC for single screening. As a result, we observed a dose-dependent robust suppression of phagocytosis by L-NE, dobutamine, and isoproterenol for 3 days (Fig. 4a), even though some showed slight changes in cell viability after 72 h of treatment (Supplementary Fig. 5). Further detailed analysis revealed the significant suppression of phagocytosis by L-NE, dobutamine, and isoproterenol, as well as by NE, but not salbutamol (Fig. 4b),

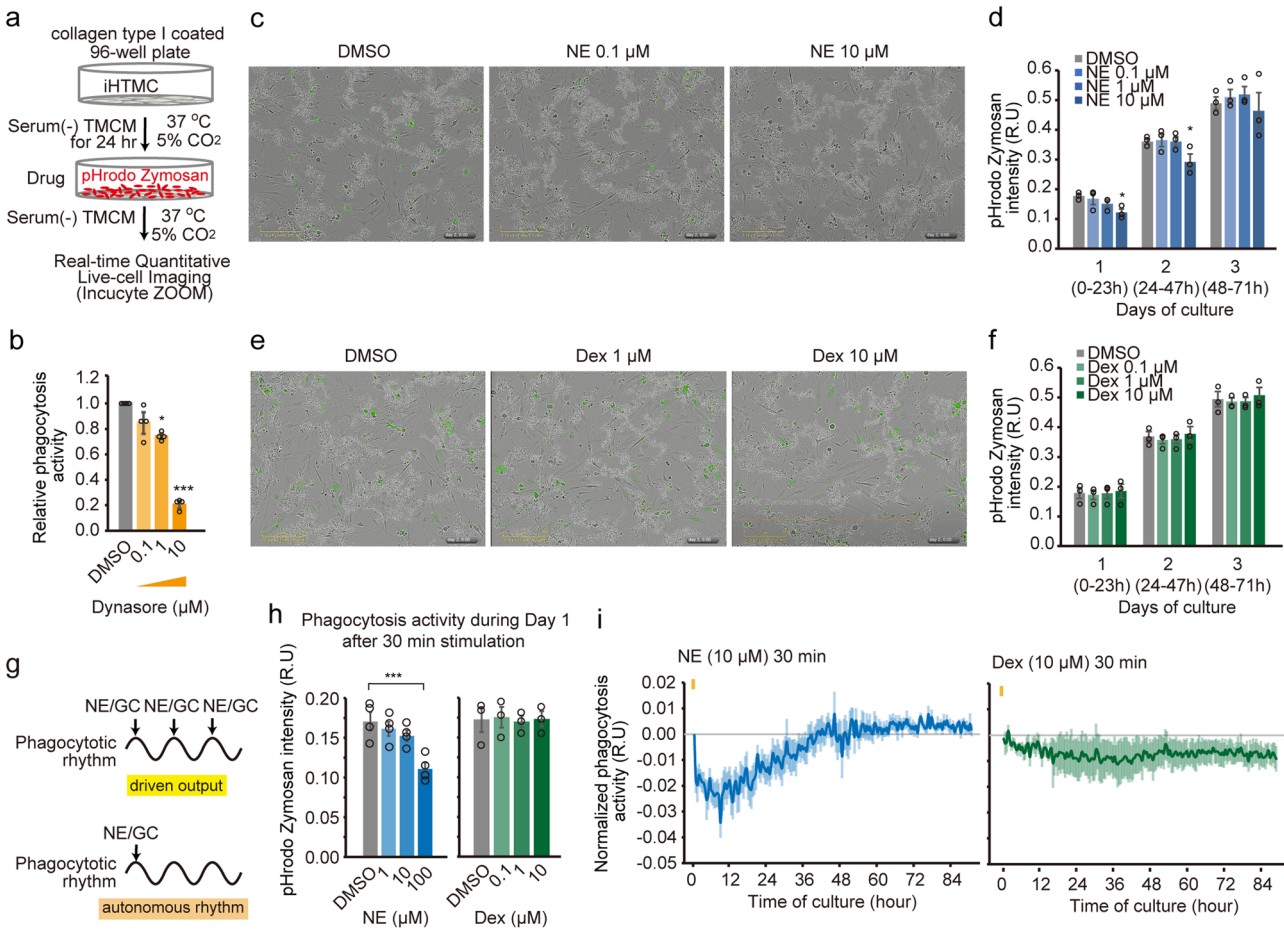

**Fig. 3 Driven-output stimuli of norepinephrine may suppress phagocytosis in iHTMC. a** Immortalized human TM cells (iHTMC) were placed on type I collagen-coated plates, and after 24 h, pH-sensitive pHrodo Zymosan was added with norepinephrine (NE) and dexamethasone (Dex) to evaluate phagocytosis in real time over 3 days. **b** This ability was confirmed by the phagocytosis inhibitor, Dynasore (*$p < 0.05$, ***$p < 0.001$ vs. DMSO, one-way ANOVA [$p < 0.001$], Dunnett's multiple comparison). Data are presented as scatter plots with mean ± SEM ($n = 3$ independent experiments). Representative images 24 h after medium changes including NE or Dex (0.1, 1, and 10 μM) (**c, e**), and quantified phagocytic activity (**d, f**). Scale bar = 300 μm. Continuous NE exposure dose-dependently prevented phagocytic activity in iHTMC during 2 days of culture (*$p < 0.05$ vs. DMSO, one-way ANOVA [$p < 0.01$], Dunnett's multiple comparison) (**d**), but not Dex (**f**). Data **b, d, f** are presented as scatter plots with mean ± SEM ($n = 3$ independent experiments). **g** driven output regulation, or self-sustainable regulation after a single stimulation in diurnal phagocytosis changes. **h, i** Exposure to NE (1, 10, and 100 μM) or Dex (0.1, 1, and 10 μM) stimulation for 30 min to iHTMC and real-time phagocytic activity was observed in iHTMC for 3 days. Although we did not observe any circadian phagocytosis rhythm in the normalized fluorescence signal, a high NE pulse stimulation (yellow bar; 10 μM) suppressed phagocytosis for about 1 day (***$p < 0.001$ vs. DMSO, one-way ANOVA [$p < 0.001$], Dunnett's multiple comparison) (**h, i**), which was later restored to normal level (**i**), but Dex pulse stimulation (yellow bar; 10 μM), did not modulate phagocytosis rhythm (**h, i**). Data are presented as bar graphs (mean ± SEM) with scatter dot plots (**h**) and mean ± SEM (**i**) of independent experiments ($n = 3$–4).

indicating the involvement of β1-AR in phagocytosis suppression. Furthermore, the AR antagonists timolol (β1β2), betaxolol (β1), and ICI-118.551 (β2) tended to rescue NE-suppressed phagocytosis but not phentolamine (α-AR antagonist) in iHTMC, whereas antagonist alone showed no effect (Fig. 4c), indicating the involvement of β1β2-AR in phagocytosis suppression. To clarify this, we analyzed the effect of RNA interference on β1β2-AR and iHTMC phagocytosis, using siRNA against *ADRB1* and *ADRB2* (Fig. 4d), after verifying the inhibitory effect on gene expression (Supplementary Fig. 6a). *ADRB1* siRNA significantly suppressed ADRB1 protein levels (Supplementary Fig. 6b) and increased NE-reduced phagocytic activity, but not *ADRB2* siRNA (Fig. 4e). This difference may be due to the conversion of Gαs to Gαi by PKA activation in β2-AR in HEK293 cells[44]. Taken together, these results present clear evidence that β1-AR mainly mediates NE effects in phagocytosis.

**β1-AR attenuates phagocytosis via the cAMP-EPAC pathway.** β1-AR is a typical G protein-coupled receptor (GPCR) that preferentially binds to stimulatory G protein $G_s$ to induce cyclic adenosine monophosphate (cAMP) production. After dobutamine stimulation in iHTMC, we detected the downstream phosphorylation of cAMP response element binding protein (CREB), but not other Gq downstream $Ca^{2+}$/calmodulin-dependent protein kinase II (CaMKII), and Gi/Gq downstream protein kinase C (Fig. 5a–c and Supplementary Fig. 7), indicating the activation of the Gs-coupled GPCR. In fact, prostaglandin E2 (PGE2), which activates Gs-coupled GPCR (EP4 receptor), prevented iHTMC phagocytosis in a dose-dependent manner (Fig. 5d). In addition, dose-dependent intracellular cAMP accumulation by cAMP inducers forskolin (FSK; Fig. 5e) and β1-AR agonist (L-NE and dobutamine; Fig. 5f) were observed, while betaxolol suppressed dobutamine-induced cAMP accumulation

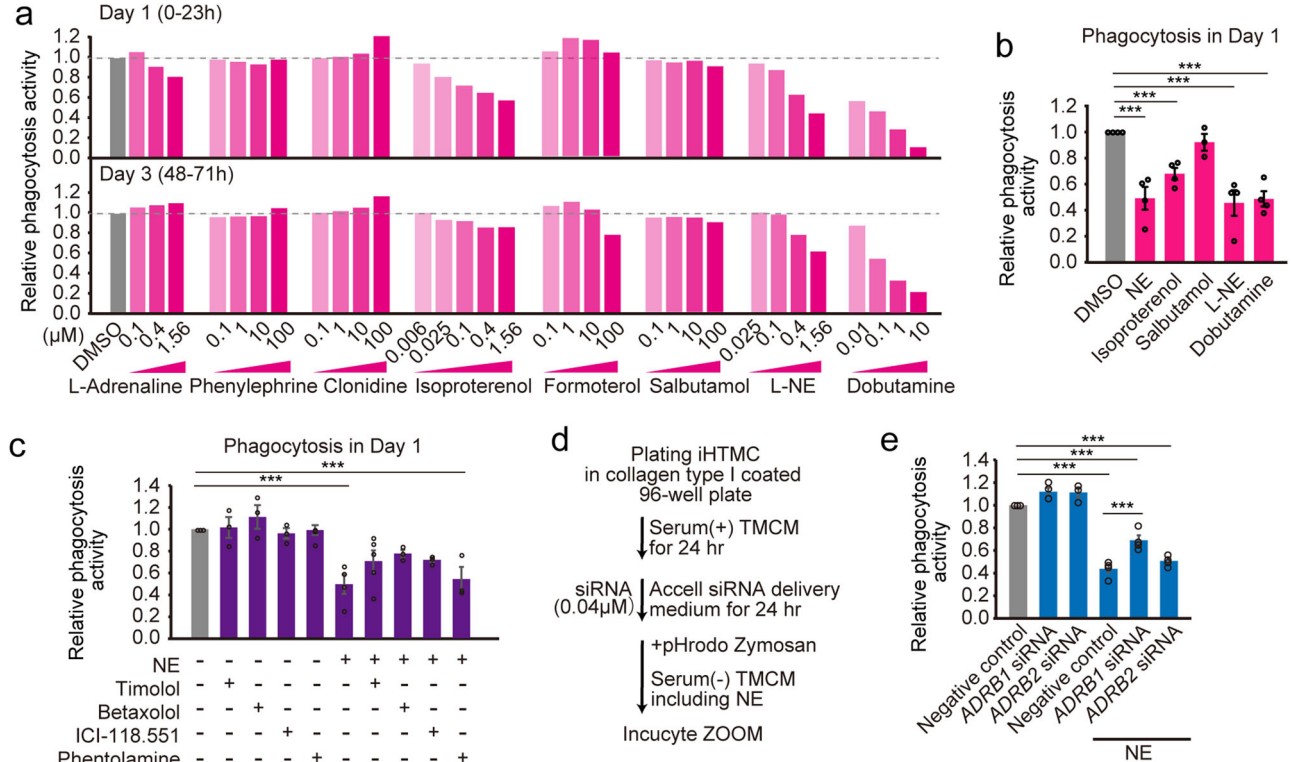

**Fig. 4 β1-adrenergic receptor (AR) mainly attenuated phagocytosis in iHTMC. a** Effects of adrenergic receptor (AR) agonists on immortalized human TM cell (iHTMC) phagocytosis for 3 days. Data are presented as the mean ($n = 1$; quadruple experiment). Phagocytosis activities were normalized to the control DMSO. Dose-dependent robust suppression of phagocytosis by AR-agonist L-adrenaline, β1-AR agonists (L-NE and dobutamine), and β1β2-AR agonist (isoproterenol) for 3 days, and no effect of selective α1-AR agonist (phenylephrine), direct-acting α2-AR agonist (clonidine), and β2-AR agonists (formoterol and salbutamol). **b** Detailed analysis revealed the significant suppression of phagocytosis by L-NE, dobutamine, and isoproterenol as well as NE, but not by the β2-AR agonist salbutamol during day 1 (***$p < 0.001$ vs. DMSO, one-way ANOVA [$p < 0.001$], Dunnett's multiple comparison). Data are presented as bar graphs (mean ± SEM) with scatter dot plots of independent experiments ($n = 4$). **c** AR antagonists timolol (β1β2), betaxolol (β1), and ICI-118.551 (β2) simultaneously with NE (5 µM) tended to rescue NE-suppressed normalized phagocytic activity but not phentolamine (α-AR antagonist) in iHTMC during day 1 (*$p < 0.05$, ***$p < 0.001$, one-way ANOVA [$p < 0.001$], Tukey's multiple comparison). Data are presented as bar graphs (mean ± SEM) with scatter dot plots of independent experiments ($n = 3-5$). **d, e** Effect of RNA interference on β1β2-AR by siRNA against ADRB1 and ADRB2 on iHTMC phagocytosis. **d** After 24 h siRNA (finally 0.04 µM) exposure with accell siRNA delivery medium, media were changed to those containing NE (1 µM). **e** ADRB1 siRNA significantly increased phagocytic activity, but not ADRB2 siRNA (***$p < 0.001$, one-way ANOVA [$p < 0.01$], Tukey's multiple comparison). Data are presented as bar graphs (mean ± SEM) with scatter dot plots of independent experiments ($n = 3-4$).

(Fig. 5g). These data demonstrate the importance of functional Gs-coupled β1-AR in HTMC phagocytosis.

cAMP binds to activate protein kinase A (PKA) or exchange proteins directly activated by cAMP (EPACs)[45]. In microglia and peritoneal macrophages, myelin phagocytosis occurs with the involvement of both EPAC1 and PKA[46]. iHTMC phagocytic activity was slightly but significantly reduced by cAMP inducers (FSK and 3-Isobutyl-1-methylxanthine [IBMX]) and cAMP analogs, such as the PKA activator Sp-cAMP and the EPAC/PKA activator 8-CPT-cAMP (Fig. 5h). To clarify the importance of PKA and EPAC in phagocytosis, RNA interference against *PRKACA*, *RAPGEF3*, and *RAPGEF4*, encoding PKA, EPAC1, and EPAC2, respectively, rescued β1-AR-suppressed iHTMC phagocytosis to a control level ($p > 0.05$, vs. control; Fig. 5i). Furthermore, blocking of PKA and EPAC1/2 by antagonists (KT5720 and ESI09, respectively) prevents β2-AR- and PGE2-suppressed neutrophil phagocytosis[47], which dose-dependently rescued dobutamine-mediated phagocytosis inhibition (Fig. 5j), indicating the involvement of both pathways. NE and isoproterenol suppress the phagocytosis of microglia cells via EPAC activation[48], while these findings clearly suggest that Gs-coupled β1-AR modulates iHTMC phagocytosis via PKA and EPAC.

**NE decrease PIP3 through SHIP activation via EPAC and PKA.** PI(3,4,5)P3 (PIP3) is necessary for phagocytosis[49] (Fig. 6a). To verify the role of PIP3 in iHTMC phagocytosis, the PIP3 antagonist, PITenin-7, dose-dependently and significantly suppressed the phagocytic activity of iHTMC (Fig. 6b). PI3K produces PIP3 from PI(4,5)P2[50] (Fig. 6a). Class I PI3K has four isoforms: α, β, γ, and δ. PI3Kγ, but not α, β, and δ, modulates the phagocytosis of microglia, which is suppressed by cAMP-mediated EPAC activation[51]. When we exposed LY294002 (broad-spectrum inhibitor of PI3Kαβδ) and CAY10505 (selective PI3Kγ inhibitor) to iHTMC, only CAY10505 slightly but significantly restrained phagocytosis (Fig. 6c). Furthermore, we detected a decrease in PIP3 levels following dobutamine treatment (Fig. 6d). These results clarified that β1-AR-mediated PIP3 modulates phagocytosis in iHTMCs.

In macrophages, PKA does not inhibit phagocytosis, whereas Epac1 exerts an inhibitory effect mainly through the activation of the tyrosine phosphatase SHIP1[52] (Fig. 6). SHIP1 converts PIP3 to PI(3, 5)P2[50] (Fig. 6a). In addition, activated SHIP1 has been reported to dephosphorylate PTEN, catalyzing PIP3 in macrophages[52], and single-cell RNA-seq analysis has suggested that both SHIP1 and PTEN are expressed in human TM

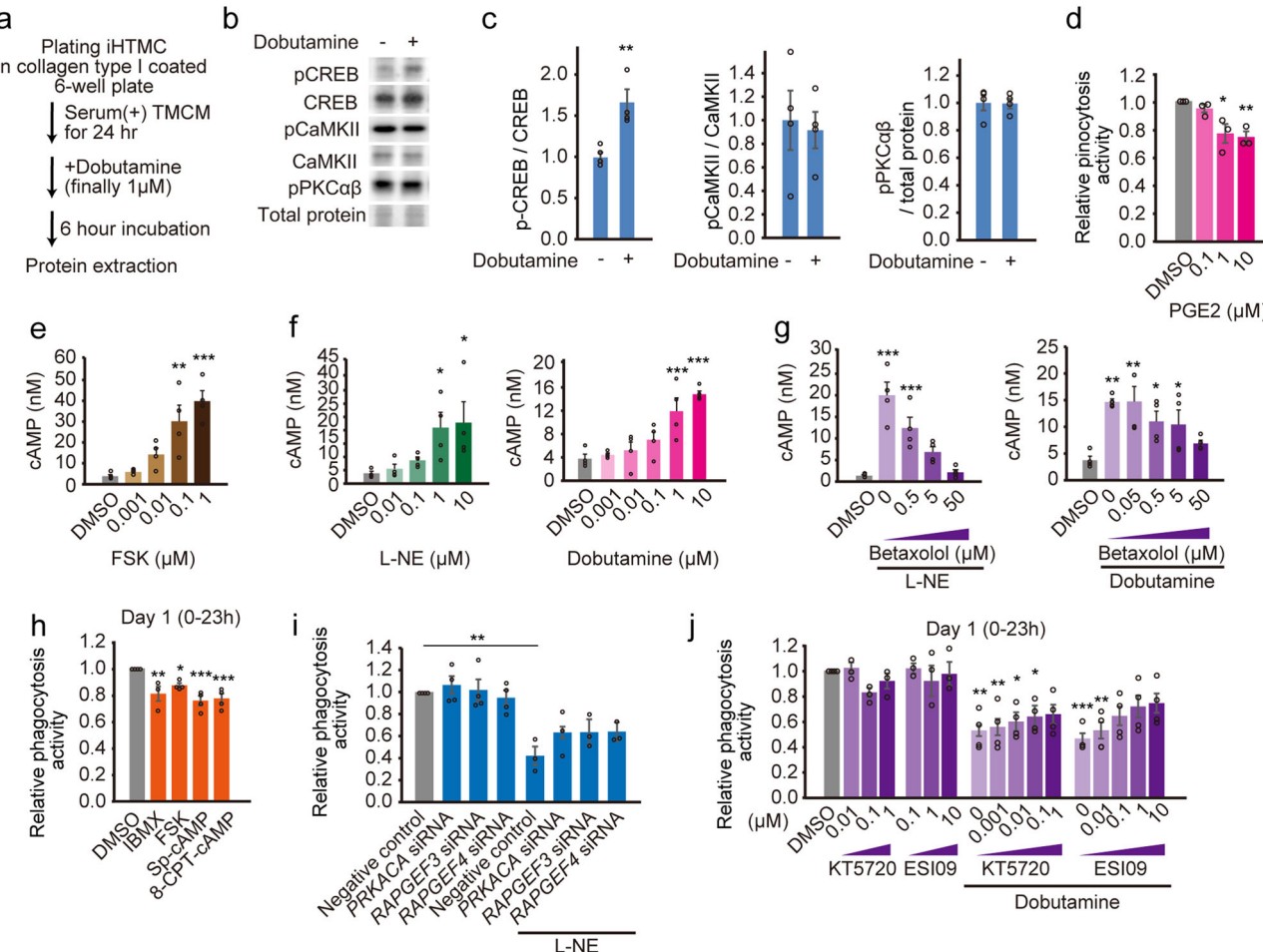

**Fig. 5 β1-AR mainly attenuated phagocytosis through cAMP-EPAC pathway. a–c** Effect of 6-h dobutamine (1 μM) stimulation on GPCR-related signaling pathways. **b, c** Western blot analysis revealed that phosphorylated CREB (pCREB) (Gαi/Gαs-related marker) was increased by dobutamine (t-test, **$p < 0.01$), but not phospho-CaMKII alpha (pCaMKII) (Gαq-related maker) and phospho-PKCα/βII [pPKCα/β] (Gαi/Gαq). Data are normalized by unphosphorylated protein or total protein and are presented as bar graphs (mean ± SEM) with scatter dot plots of independent experiments ($n = 4$). **d** Prostaglandin E2α (PGE2α), activating Gs-coupled GPCR, dose-dependently prevented iHTMC phagocytosis (*$p < 0.05$, **$p < 0.01$ vs. DMSO, one-way ANOVA [$p < 0.001$], Dunnett's multiple comparison). Data are presented as bar graphs (mean ± SEM) with scatter dot plots of independent experiments ($n = 3$). Phagocytosis activity was normalized to that of control DMSO. Dose-dependent intracellular cAMP accumulation by cAMP inducers forskolin (FSK) (**e**), and by L-NE and dobutamine (**f**) (*$p < 0.05$, **$p < 0.01$, ***$p < 0.001$ vs. DMSO, one-way ANOVA [$p < 0.001$], Dunnett's multiple comparison). **g** betaxolol dose-dependently suppressed L-NE (10 μM)- or dobutamine (1 μM)-induced cAMP accumulation (*$p < 0.05$, **$p < 0.01$ vs. DMSO, one-way ANOVA [$p < 0.001$], Dunnett's multiple comparison). Data **e–g** are presented as bar graphs (mean + SEM) with scatter dot plots of independent experiments ($n = 4$). **h** iHTMC phagocytic activity was slightly but significantly reduced by cAMP inducers (FSK and IBMX) and cAMP analogs, such as PKA activator Sp-cAMP and EPAC/PKA activator 8-CPT-cAMP (*$p < 0.05$, **$p < 0.01$, ***$p < 0.001$ vs. DMSO, one-way ANOVA [$p < 0.001$], Dunnett's multiple comparison). Phagocytosis activity was normalized to that of control DMSO. **i** siRNA (PRKACA, RAPGEF3, and RAPGEF4) administration significantly rescued β1-AR-mediated suppression of iHTMC phagocytosis (**$p < 0.01$, one-way ANOVA [$p < 0.001$], Tukey's multiple comparison). **j** The recovery effect of PKA and EPAC1/2 antagonists (KT5720 and ESI09, respectively) simultaneously added with dobutamine (1 μM) dose-dependently rescued dobutamine-suppressed iHTMC phagocytosis (*$p < 0.05$, **$p < 0.01$, ***$p < 0.001$ vs. DMSO, one-way ANOVA [$p < 0.001$], Tukey's multiple comparison). The antagonists were added at the same time as dobutamine. Data **h–j** are presented as bar graphs (mean ± SEM) with scatter dot plots of independent experiments ($n = 3$–4).

macrophages[43] (Supplementary Fig. 8). In fact, exposure of iHTMC with PKA and EPAC inhibitors improved the PIP3 suppression efficacy of dobutamine (Fig. 6d). In addition, the PTEN inhibitor bisperoxovanadium (pyridine-2-carboxyl) [bpV(pic)][53], SHIP1 inhibitor 3-a-aminocholestane (3AC)[54] and K118[55] also improved this efficiency (Fig. 6d), indicating the involvement of the β1-AR-signaling pathway in PIP3 reduction. Based on the results presented above, we next performed western blot analysis of SHIP1 phosphorylation to determine the effects of β1-AR-mediated PKA and EPAC signaling in iHTMCs. Dobutamine induced the phosphorylation of SHIP1, which was prevented by EPAC inhibition, but not by PKA inhibition

(Fig. 5e), indicating the importance of β1-AR-mediated SHIP1 activation through EPAC.

As PIP3 stimulates AKT and ERK1/2 signaling to modulate phagocytosis in macrophages[52], we next verified the effect of the signaling pathway of NE on AKT/ERK1/2 phosphorylation (Fig. 6a, f). Dobutamine cleanly inhibited their phosphorylation, which recovered significantly with EPAC inhibitors, but not with PKA inhibitors (Fig. 6f). AKT/ERK1/2 phosphorylation suppressed by dobutamine was upregulated by PTEN and SHIP1 inhibitors (Fig. 6f). These results indicate the involvement of SHIP/PTEN-reduced PIP3 in the EPAC-mediated suppression of AKT/ERK activation. Furthermore, when we monitored the effect

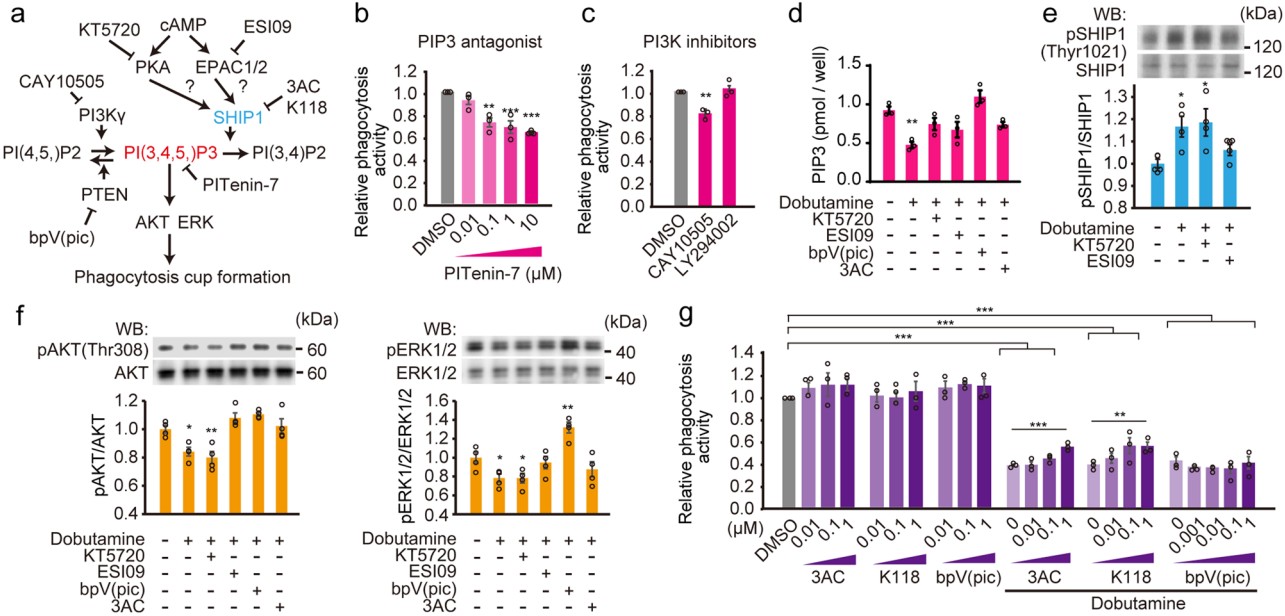

**Fig. 6 NE decreases PIP3 through SHIP activation via EPAC and PKA. a** The hypothesized phagocytosis suppression mechanism of β1-AR-cAMP. PIP3 is a hub molecule that induces phagocytic cup formation through phosphorylation of AKT and ERK, and PI3K, PTEN, and SHIP1 regulate PIP3 levels. However, the involvement of EPAC and PKA in iHTMCs in this system is unknown. **b** Administration of the PIP3 antagonist PITenin-7 dose-dependently and significantly suppressed the phagocytic activity of iHTMC (**$p < 0.01$, ***$p < 0.001$ vs. DMSO, one-way ANOVA [$p < 0.001$], Dunnett's multiple comparison). **c** Effect of LY294002 (inhibitor of PI3Kαβδ) and CAY10505 (selective PI3Kγ inhibitor) administration on phagocytosis (**$p < 0.01$, DMSO, one-way ANOVA [$p < 0.01$], Dunnett's multiple comparison). **d** PIP3 levels after 6 h exposure to dobutamine with several antagonists in iHTMC. Dobutamine (1 μM) decreased PIP3, which was improved by KT5720 and ESI09. In addition, bpV (pic) (PTEN inhibitor) and 3AC (SHIP1 inhibitor) also improved it (**$p < 0.01$, vs. control, one-way ANOVA [$p < 0.001$], Dunnett's multiple comparison). Data **b–d** are presented as bar graphs (mean ± SEM) with scatter dot plots of independent experiments ($n = 3$). **e** Western blot analysis of SHIP1 phosphorylation in iHTMCs. Dobutamine induced phosphorylation of SHIP1, which was prevented by ESI09 but not by KT5720 (**$p < 0.01$, ***$p < 0.001$ vs. DMSO, one-way ANOVA [$p < 0.001$], Dunnett's multiple comparison). Data are normalized to unphosphorylated SHIP1 and are presented as bar graphs (mean ± SEM) with scatter dot plots of independent experiments ($n = 4$). **f** Involvement of the β1-AR signaling pathway in AKT or ERK1/2 phosphorylation. Dobutamine inhibited this phosphorylation, which was recovered significantly by ESI09, bpV (pic), and 3AC, but not by KT5720 (*$p < 0.05$, **$p < 0.01$, vs. control, one-way ANOVA [$p < 0.001$], Dunnett's multiple comparison). Data are normalized by unphosphorylated proteins and are presented as bar graphs (mean ± SEM) with scatter dot plots of independent experiments ($n = 4$). **g** recovery efficacy of bpV(pic), K118 (SHIP1 inhibitor), or 3AC on dobutamine (1 μM)-mediated suppression of phagocytosis (**$p < 0.01$, ***$p < 0.001$, one-way ANOVA [$p < 0.001$], Tukey's multiple comparison). Phagocytosis activity was normalized to that in the control. Data are presented as bar graphs (mean ± SEM) with scatter dot plots of independent experiments ($n = 3$).

of PTEN and SHIP1 inhibitors on the phagocytosis suppression effect of dobutamine, only SHIP1 inhibitors dose-dependently rescued it up to the near control level, but no effect was seen for the PTEN inhibitor or the antagonist alone (Fig. 6g). These results suggest that, at least in vitro, β1-AR-mediated SHIP1 activation through EPAC reduces PIP3 to suppress phagocytic cup formation.

**Nocturnal activation of the β1-AR-EPAC-SHIP1 pathway enhanced IOP in mice.** Based on the results presented above, we next performed immunohistochemical analysis to determine the localization of Adrb1 and Ship1 in the mouse eye (Supplementary Fig. 9). This demonstrated the colocalization of Adrb1 and Ship1 within the endothelial cells in the SC and the NPE cells in the ciliary body (Supplementary Fig. 9), providing evidence of the existence of the β1-AR-EPAC-Ship1 system in mice.

Since NE in AH released from the SCG has a circadian rhythm with nocturnal increase in rodents[56], which seems to also be true in humans, a nocturnal increase of NE in TM may cause IOP increase by inhibiting phagocytosis in the TM. To validate this hypothesis, we administered mice β1-AR-SHIP1 pathway inhibitors by instillation at ZT10 and measured IOP at night (ZT15) (Fig. 7a, b). First, the β1-AR antagonist betaxolol is well used for glaucoma therapy[57], lowers topically mouse IOP[58], it blocked the nocturnal IOP increase (Fig. 7a). Furthermore, the effects of

KT5720, ESI09, and bpV(pic) on retinal photoreceptor death in rodents[59,60] have been reported but not on IOP, while that of the 3AC, promoting phagocytosis[54], on IOP in animals remains unknown. Interestingly, instillation of ESI09 and 3AC among these reagents significantly suppressed the nocturnal IOP increase (Fig. 7a) and also at the individual level ($p < 0.01$); however, the PKA inhibitor did not show this effect (Fig. 7b), suggesting a role for the β1-AR-EPAC-SHIP1 pathway in nocturnal IOP increase. However, since EPAC inhibitor ESI09 suppressed nocturnal IOP more than betaxolol (Fig. 7b), and *RAPGEF3* and *RAPGEF4* are expressed in human SC endothelial cells[43], it may also affect mechanisms such as cytoskeleton and cell adhesion in the TM, the giant vacuoles in the endothelial cells of the SC, or AH inflow in the NPE of the ciliary body other than TM phagocytosis because Epac1 mediates retinal neurodegeneration in mouse models of ocular hypertension[60].

As we demonstrated the possibility of a driven-output phagocytic activity (Fig. 3), we next measured day-night changes in IOP 1 day before and after instillation to verify whether SHIP1 inhibition suppresses IOP rhythm in a self-sustainable manner (Fig. 7c). Interestingly, SHIP1 inhibition by single instillation did not show any inhibitory effect on nocturnal IOP increase the day after instillation (Fig. 7c), providing evidence of driven-output IOP enhancement by nocturnal NE (Fig. 7d). This regulatory pathway is consistent with β1-AR-mediated circadian regulation

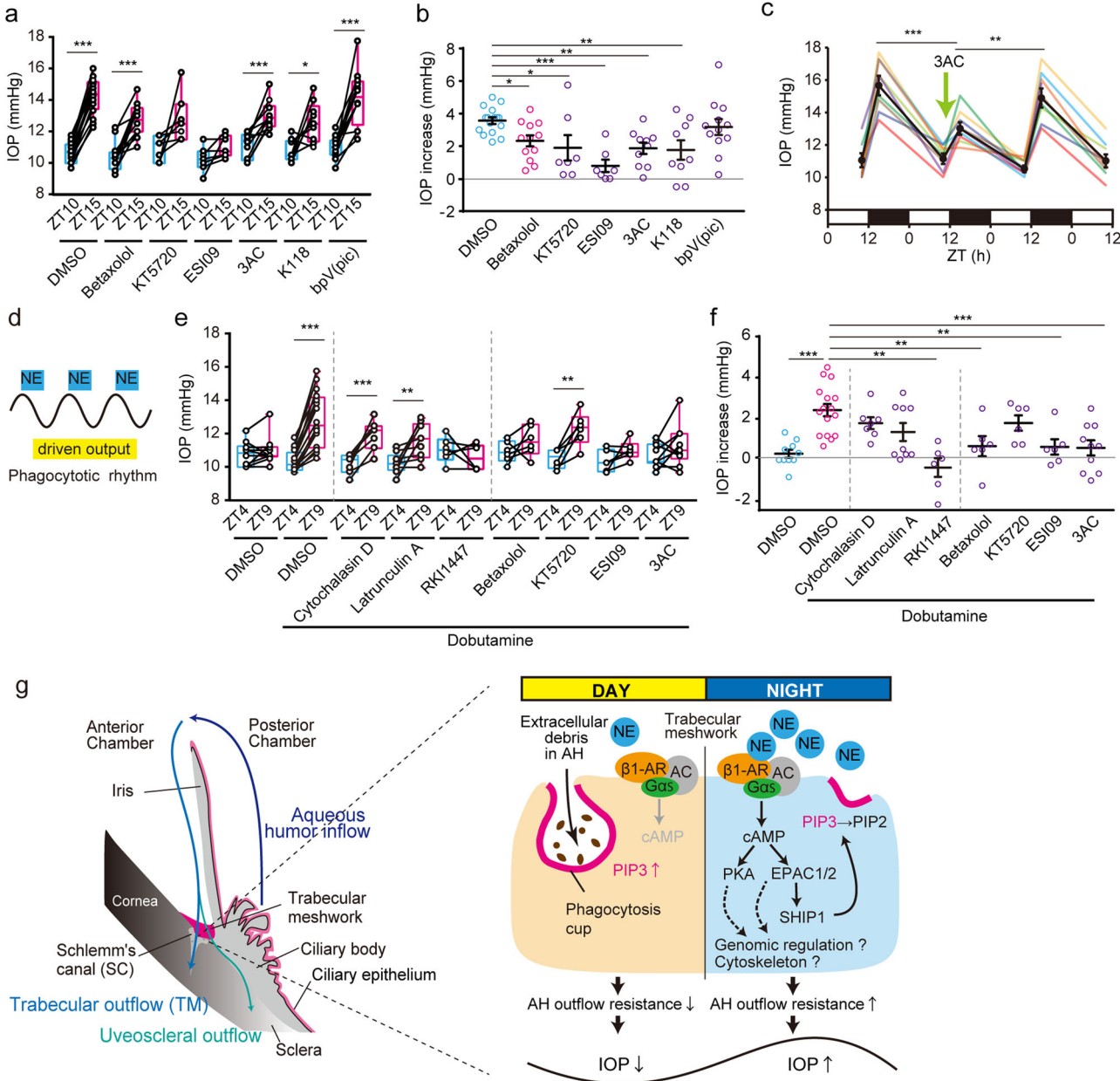

**Fig. 7 Nocturnal activation of the β1-AR-EPAC-SHIP1 pathway enhances IOP in mice. a, b** Effect of β1-AR-EPAC-SHIP1 pathway-related inhibitors on nocturnal IOP increase in mice. Mice were administered betaxolol (100 μM/0.1% DMSO PBS), KT5720 (100 μM), ESI09 (100 μM), 3AC (1 mM), K118 (1 mM), and bpV(pic) (100 μM) at ZT10, and IOP was measured at ZT15. **a** Instillation of KT5720 and ESI09 significantly suppressed the nocturnal IOP increase (paired t-test, **$p < 0.01$, ***$p < 0.001$). Data are presented as box-and-whisker plots with individual data ($n = 7$–16). **b** All inhibitors except bpV (pic) prevented an IOP increase (*$p < 0.05$, **$p < 0.01$, ***$p < 0.001$ vs. DMSO, one-way ANOVA [$p < 0.001$], Dunnett's multiple comparison test). Data are presented as scatter plots with mean ± SEM ($n = 7$–16). **c** Day-night changes in IOP one day before and after 3AC instillation. Data are presented as mean ± SEM ($n = 10$) with individual data. One day after single instillation, IOP showed a normal nocturnal increase (**$p < 0.01$, ***$p < 0.001$, one-way ANOVA [$p < 0.001$], Bonferroni's multiple comparison test), providing evidence of driven-output IOP enhancement by nocturnal NE (**d**). **e, f** Validation of IOP inhibitory effects of drugs on β1-AR-mediated IOP regulation using instillation of dobutamine (100 μM) at ZT4. After 5 h, the IOP was measured. Dobutamine significantly enhanced IOP, while pretreatment with betaxolol, ESI09, and 3AC prevented this effect (paired t-test, **$p < 0.01$, ***$p < 0.001$). **f** The same was true for individual data (**$p < 0.01$, ***$p < 0.001$ vs. Dobutamine+DMSO, one-way ANOVA [$p < 0.001$], Dunnett's multiple comparison test). Data are presented as **e** box-and-whisker plots with individual data and **f** as scatter plots with mean ± SEM ($n = 6$–16). **g** Regulatory model of time-dependent systems in which nocturnal NE suppresses phagocytosis to induce AH outflow resistance through β1-AR-EPAC-SHIP1 activation, leading to an IOP increase at night. The AH produced from the ciliary body is drained by two pathways: trabecular outflow and uveoscleral outflow.

of the pineal gland[61]. Since NE from the SCG appears to show a nocturnal peak in rodents[42], the circadian rhythm of SCG-NE can regulate diurnal changes in TM phagocytosis (Fig. 7d).

To verify the IOP inhibitory effects of drugs on β1-AR-mediated IOP regulation in vivo, we administered dobutamine

instillation at ZT4 to increase IOP (Fig. 7e). Five hours after administration, IOP was significantly and clearly enhanced (Fig. 7e, f). To clarify the involvement of cytoskeletal rearrangement and phagocytosis in this enhancement, we next evaluated cytochalasin D, latrunculin A, and RKI1447. Interestingly, only

RKI1447, a phagocytosis promoter, prevented this increase (Fig. 7e, f), indicating the contribution of TM phagocytosis in β1-AR-mediated IOP increase rather than actin rearrangement. In other words, the decrease in AH outflow by phagocytosis suppression outweighed that of actin rearrangement in the other TM cells. Conversely, since these inhibitors suppressed nocturnal IOP increase (Fig. 2), the increase in AH outflow by actin depolymerization may increase at night. In addition, importantly, preinstillation with a β1-AR antagonist prevented this increase (Fig. 7e, f), confirming the effect of β1-AR. To determine the role of EPAC and SHIP1 in this effect, we blocked dobutamine-activated PKA, EPAC, or SHIP1 (Fig. 7e, f). Preinstillation of EPAC and SHIP1 inhibitors significantly suppressed IOP increase (Fig. 7e, f) and prevented IOP increase in individual data ($p < 0.01$); however, the PKA inhibitor did not show this effect (Fig. 7f), indicating the importance of the β1-AR-EPAC-SHIP1 pathway in IOP regulation. Taken together, these findings revealed that nocturnal NE suppresses phagocytosis-mediated AH outflow through β1-AR-EPAC-SHIP1 activation, leading to an increase in IOP at night (Fig. 6g).

## Discussion

Previous studies have indicated that sympathetic NE and adrenal GC transmit circadian timing signals to the eye to generate IOP[13]. However, the involvement of NE and GC in TM phagocytosis contributing to IOP regulation remains unknown[62,63]. In this study, we found that the suppression of AH drainage in mice may partially contribute to the increased nocturnal IOP. In addition, we demonstrated that driven-out suppressed TM phagocytosis by NE in vitro, which is regulated by β1-AR-cAMP-EPAC-SHIP1 activation. Although NE is known to suppress macrophage phagocytosis[26], and PIP3 stimulates AKT and ERK1/2 signaling to modulate phagocytosis[49,52], the connection between NE and PIP3-triggered phagocytosis remains unclear. In this study, we identified β1-AR -suppressed TM phagocytosis by silencing PIP3-AKT or -ERK signaling through the cAMP-EPAC-SHIP1 pathway. In the TM of the POAG donors, the cAMP signaling pathway and CREB were activated[64], while ERK phosphatase activity was downregulated[65]. These results support those of the present study. Further understanding of this pathway is necessary to fully explain the complex cellular mechanisms by which occupancy of specific ARs regulates AH dynamics, which may contribute to the establishment of chronotherapy.

In the present study, the time-dependent mechanism of the inhibitory effect of NE on TM phagocytosis was elucidated, but the circadian regulation of NE/GC in other important key factors such as cytoskeleton, cell adhesion, and IOP-independent uveoscleral outflow to the ciliary muscle (Fig. 6i) in the AH outflow remain unknown. Long-term stimulation of GC increases the actin polymerization of TM and induces fibrosis[66]. In the TM, cAMP/PKA activation and downstream RhoA inactivation lead to a loss of actin stress fibers and focal adhesions and disassembly of the matrix network[67]. Ship1−/− in neutrophils upregulates basal actin polymerization[68]. However, the time-dependent efficacy may explain their contribution to some extent. The effects of actin polymerization inhibitors on IOP reduction seem to be limited to the dark period in mice (Figs. 2, 7). In fact, the decrease in AH outflow by phagocytosis suppression on day appears to outweigh that by actin rearrangement in the TM. The cell-intrinsic circadian clock regulates numerous cytoskeletal regulators in fibroblast[69]. In contrast, in mice, 30–42% of AH passes through the uveoscleral pathway[70]. To understand the AH dynamic rhythm completely, these determinants need to be elucidated. The discovery of a phagocytosis activator independent of the cytoskeleton or single-cell

analysis separated by the function of TM will be able to elucidate them completely in the future.

In general, β-AR blockers, including betaxolol, effectively reduce IOP by decreasing AH inflow in patients[71], while several studies have demonstrated the opposite effect of the sympathetic role in AH outflow[72,73]. β-AR may be involved in the increase of AH outflow by reducing the size of cells in the TM[72]. Continuous electrical stimulation of the cervical sympathetic nerve decreased IOP only during 1 h[73]. However, these studies demonstrated the role of β2-AR. In fact, POAG involves a greater increase in IOP at night when SCG-NE release is thought to be higher than during day[10]. Furthermore, a recent study revealed that betaxolol is the top compound most opposed to POAG signatures calculated by microarray database analysis[74]. β1-AR may contribute to nocturnal AH outflow resistance.

Since, in a previous study demonstrating that the double knockout mouse model of β1 and β2 ARs maintains IOP rhythm[7], these mice had different genetic backgrounds compared with the control mice, we think it is impossible to assess the IOP rhythmicity such as its amplitude. However, it is certain that β1/2-ARs are not essential for IOP rhythm formation[7]. In the present study, betaxolol did not completely prevent nocturnal IOP rise (Fig. 7). The removal of SCG in mice attenuated the IOP rhythm[13], and betaxolol instillation resulted in IOP rhythms with nocturnal peak flattening in both POAG and NTG patients[57]. The contribution of the β1-AR-mediated circadian rhythm of AH resistance for IOP rhythm formation may not be so high, and Dex or the other AR can regulate AH-outflow/inflow. β2-AR- or GC-mediated AH production in the NPE of the ciliary body is related to the IOP rhythm[13]. This may explain why β-AR1/2 double knockout mice maintain this IOP rhythm.

In the present study, we found that NE was a necessary condition for GC-mediated phagocytosis in the TM. Both β-adrenergic signaling and GCs are mediators of SCN timing signals in osteoblasts[75]. Interactions between the sympathetic nervous system and GCs have also been previously reported. In particular, GC transcriptionally modulates β2-AR expression by modulating GC-response elements on the promoter[76]. Interestingly, GCs rapidly activate cAMP production via Gαs to initiate non-genomic signaling, which contributes to one-third of their canonical genomic effects[77]. In fact, betaxolol prevented steroid-induced IOP increase[78]. Thus, in TM, Gs-bound GR may enhance the β1-AR-Gs signal to suppress phagocytosis and generate an appropriate AH drainage rhythm.

Our study has several inherent limitations. First, although we provide a model of IOP induction in nocturnal NE, the circadian rhythm of NE released from SCG in humans remains unclear. Although GC secretion peaks at the light offset[79], and NE is released with a nocturnal peak from the SCG in rodents[42], GC rhythms are anti-phasic but not SCG-NE in diurnal animals[42,80]. Nocturnal NE release from SCG generally stimulates melatonin synthesis in humans, as well as in other mammals[81]. In humans, β-blockers inhibit nocturnal melatonin levels[82], and suppress IOP increase during late night to morning[83]. Furthermore, in nocturnal rabbits, β-blockers suppress IOP increase only at night[84]. These results indicate the involvement of nocturnal NE release from the SCG in regulating the IOP rhythm. Second, we cannot fully explain the differences in IOP rhythms in diurnal and nocturnal animals using the present model. The IOP rhythm peaks early at night in nocturnal animals[7], while in healthy humans, it appears to be elevated during the night and peaks from late night to early morning[6,9]. In phasic SCG-NE and anti-phasic GC, the action of both factors in IOP increase may explain such differences. Third, protein downregulation by RNA interference in the present study might not be sufficient for the contribution of target proteins, so knock-out analysis in vitro or

tissue-specific gene modification techniques using virus in vivo can help the understanding.

Taken together, these results suggest a potential circadian role for NE in the modulation of phagocytosis and AH outflow resistance by TM, contributing to the IOP rhythm. Although the TM accounts for most outflow and is the major site of AH outflow resistance[70,85], until the approval of the ROCK inhibitor, no clinically administered drug had a direct effect on the TM. In the present study, we suggest the possibility of therapeutic drug development targeting TM. In addition, the synergistic regulatory mechanism of GC in the presence or absence of NE in AH dynamics remains unknown. Since genomic signaling may also contribute to this mechanism[77] (Fig. 7g), further understanding of the time-dependent efficacy of GC and NE on AH inflow/outflow will lead to a complete elucidation of the regulatory mechanisms of IOP rhythm. Although new therapeutics with new mechanisms, such as chronotherapy, are urgently needed for glaucoma treatment, the development of multiple types of drugs using this interaction could be especially useful for glaucoma treatment in the future.

## Methods

**Animals**. Five-week-old male C57BL/6JJmsSlc mice ($N = 130$; Japan SLC Inc., Shizuoka, Japan) were purchased and housed in plastic cages (170 W × 240 D × 125 H mm; Clea, Tokyo, Japan) under a 12 h light (200 lx of fluorescent light)/dark cycle (12L12D, 0800 light ON, 2000 light OFF), maintained at a constant temperature (23 ± 1 °C). Food (CE-2; CLEA) and water were provided *ad libitum*[13]. All animal experiments were approved by the Committee of Animal Care and Use of the Aichi Medical University. All experimental procedures were conducted in accordance with the institutional guidelines for the use of experimental animals[13].

**IOP measurement**. IOP measurements were performed using a tonometer (Icare TonoLab, TV02; Icare Finland Oy, Espmoo, Finland), as previously reported[8,13]. All mice were kept under 12L12D conditions for more than 2 weeks before IOP measurements. Unanesthetized mice were gently held using a sponge. IOPs were measured during the light phase under light (200 lx) conditions and during the dark phase under dim red-light conditions. For the analysis of phagocytosis-related drugs and dobutamine-mediated IOP induction, IOP was measured before drug instillation at zeitgeber time (ZT) 4 and measured at ZT9. ZT0 (0800) was defined as the time of light ON. To analyze the nocturnal IOP increase, IOP was obtained by measuring the IOP at ZT10 and ZT15. We calculated back from ZT15 (the peak of IOP) for 5 h and set it to ZT10 when IOP was low. For diurnal changes in IOP, IOP was measured at ZT6 (IOP trough) and ZT15 (IOP peak)[13], 2 weeks after bead injection.

**Intraocular injection and detection of fluorescence particles**. To investigate the effect of AH inflow on nocturnal IOP increase, mice were anesthetized by isoflurane inhalation (2%; WAKO, Saitama, Japan), supplemented with topical proparacaine HCl (0.5%; P2156, Tokyo Chemical Industry [TCI], Tokyo, Japan), and were treated with an intraocular injection of an Na$^+$/K$^+$ATPase inhibitor ouabain (100 μM/0.1% dimethyl sulfoxide [DMSO], phosphate-buffered saline [PBS], 3 μL) into anterior chamber of the right eye with a 34-gauge needle (0.18 × 8 mm, Pasny; Unisis) connected to a Hamilton syringe at ZT10. To precisely control the small volume (3 μL) of anterior chamber injection, 3 μL PBS (0.1% DMSO) was injected into the left eye with a 34-gauge needle connected to a Hamilton syringe.

For microbead injection to prevent AH outflow, mice were anesthetized by isoflurane inhalation (2%; WAKO, Japan), supplemented with topical proparacaine HCl. IOP elevation was induced unilaterally in adult C57BL/6J mice by injection of 3 μL of 1/10 diluted fluosphere polystyrene microspheres (15 μm, yellow-green fluorescent, F8844, Invitrogen, Carlsbad, CA) into the anterior chamber of the right eye with a 34-gauge needle connected to a Hamilton syringe. Microbeads were then resuspended in PBS at $5.0 \times 10^6$ beads/mL. To precisely control the small volume (3 μL) of anterior chamber injection, 3 μL PBS was injected into the left anterior chamber with a 34-gauge needle, connected to a Hamilton syringe.

To visualize the AH outflow in the SC, mice were anesthetized by isoflurane inhalation (2%; WAKO, Japan), supplemented with topical proparacaine HCl, and were treated with an intraocular injection of 3 μL of carboxylate-modified microspheres (0.5 μm, yellow-green fluorescent, 2% solids, F8813, Invitrogen) at ZT0 and ZT12. After 6 h, the injected mice were anesthetized, and their anterior eyes were extracted and fixed in 4% paraformaldehyde (26126-25, Nakarai Tesk)/PBS for 5 min. After washing with PBS, anterior eye cups were placed in Hanks' Balanced Salt Solution (HBSS; 082-08961 WAKO) in a 96-well microplate to observe the fluorescence from the bottom using a digital fluorescent microscope (Dino-Lite Edge M Fluorescence TGFBW; Opto Science Inc., Tokyo, Japan), and

the fluorescence intensity (525 nm) was measured using a microplate reader, SpectraMax M5 (Molecular Devices).

**Drug installation**. Drug instillation was performed as described in our previous report[13]. Unanesthetized 8-week old male mice were used in this study. For the analysis of phagocytosis and/or cytoskeleton-related drugs, mice were instilled with a single drop (30 μL) of Dynamin-related phagocytosis inhibitor Dynadore (1 mM/0.1% DMSO PBS; D5461, TCI), actin polymerization and phagocytosis inhibitor cytochalasin D (100 μM; 11330, Cayman), latrunculin A (100 μM; 125-04363, WAKO), and ROCK inhibitor RKI1447 (1 mM; 16278, Cayman) at ZT4 using a micropipette into bilateral eyes, and IOP was measured at ZT10. For nocturnal IOP increase analysis, mice were instilled with a single drop of Dynadore, cytochalasin D, latrunculin A, RKI1447, β1-AR antagonist betaxolol hydrochloride (100 μM/0.1% DMSO PBS; B4474, TCI), PKA inhibitor KT5720 (100 μM; 10011011, Cayman), EAPC1/2 inhibitor ESI09 (100 μM; 19130, Cayman), 3α-aminocholestane (1 mM, 3AC; HY-19776, MCE), pan-SHIP1 inhibitor K118 (1 mM; B0344, Echelon Biosciences, Salt Lake City, USA), and PTEN inhibitor bpV (pic) (100 μM; SML0885, Sigma-Aldrich) at ZT10 using a micropipette in both eyes, and IOP was measured at ZT15. To analyze the inhibitory efficacy of dobutamine-induced IOP increase, the above antagonists were preinstilled at ZT4, after 10 min, a single drop of dobutamine (100 μM/0.1% DMSO PBS) was added. During instillation, the mice were gently restrained with necks held back.

**iHTMC culture**. The immortalized human TM-SV40 cell line (iHTMC) derived from primary human SC and TM regions was purchased from Applied Biological Materials Inc. (T-371-C, ABM Inc., Richmond, BC, Canada) and cultured in TM cell medium (6591, Sciencell, Carlsbad, CA, USA) supplemented with 2% fetal bovine serum (0010, ScienCell), 1% growth supplement (TMCGS, 6592, Sciencell) and 1% penicillin/streptomycin (0503, Sciencell) in type I collagen-coated 100-mm dish (3020-100, IWAKI, Japan). Experiments were performed on type I collagen-coated plates. Upon reaching confluence, iHTMCs were split 1:4 using 0.05% trypsin/PBS. Cell viability was determined using trypan blue (0.4%) exclusion.

**Phagocytosis assay**. For phagocytosis assay, iHTMCs (T0371, abm) were plated in collagen I-coated 96-well microplates (4860-010; IWAKI) at a density of $5.0 \times 10^3$ cells/well in trabecular meshwork cell medium (TMCM) supplemented with 1% penicillin/streptomycin and growth factors (6591; Sciencell). To measure phagocytosis, pHrodo Green Zymosan Bioparticles (P35365; ThermoFisher) were suspended in TMCM and vortexed to disperse. After 90% confluence, the medium was removed by aspiration, and 100 μL of serum-free TMCM was immediately added. After 24 h, the medium was replaced with serum-free TMCM containing pHrodo Zymosan (2.5 μg/well) in the presence of several kinds of drugs, and the plate was placed in the IncuCyte ZOOM instrument (Essen Bioscience, Ann Arbor, MI, USA), installed in a 5% CO$_2$ incubator at 37 °C. Each well was imaged at 3 points, every 0.5 h or 1 h for more than 72 h using the channels of green fluorescence and phase and the 10× objective (Nikon). No pHrodo Zymosan was used for fluorescent control using background fluorescent intensity because of autofluorescence in TMCM, and vehicle control included 0.1% DMSO. The green fluorescence intensity at each time point in each well was measured using the IncuCyte ZOOM 2015A software (Essen Bioscience). To perform a detailed analysis of the pulse stimulation, we calculated the difference from the control. We calculated the average daily fluorescence intensity (0–23 h [Day 1], 24–47 h [Day 2], and 48–71 h [Day 3]) normalized to that of the control DMSO-treated group to show statistical changes. For pulse stimulation of drug, phagocytosis index is calculated by the difference with the control. The phagocytosis assay was independently repeated thrice using three or four biological replications.

**Drug treatment**. For the phagocytosis assay, iHTMCs were simultaneously treated with (−)-NE (0.1, 1, and 10 μM; S9507, Selleck) and/or dexamethasone (Dex; 0.1, 1, and 10 μM; 11107-51, Nakarai Tesk, Kyoto, Japan) and pHrodo Zymosan (2.5 μg/well), and were exposed over 3 days without washout. To confirm the phagocytic activity of iHTMCs, we treated iHTMCs with the phagocytosis inhibitor Dynasore (0.1, 1, and 10 μM; D5461, TCI). For a detailed analysis of pulse stimulation, serum-free TMCM containing NE (1, 10, and 100 μM) or Dex (0.1, 1, and 10 μM) was added to iHTMC for 30 min. After washing with TMCM, pHrodo zymosan-containing TMCM was added to the iHTMC. For analysis of effect of agonists on phagocytosis, we treated several kinds of agonists for AR agonist L-adrenaline (A0173, TCI), β1-AR agonists [L-Noradrenaline Bitartrate Monohydrate (L-NE, A0906, TCI) and dobutamine hydrochloride (15582, Cayman)], β1β2-AR agonist [Isoproterenol Hydrochloride (I0260, TCI)], selective α1-AR agonist [L-Phenylephrine (P0395, TCI)], direct-acting α2-AR agonist [Clonidine HCl (038-14291, WAKO)], β2-AR agonist [Formoterol fumarate hydrate (F0881, TCI)], short-acting β2-AR agonist [Salbutamol Hemisulfate [S0531, TCI]], PGE2 (0.1, 1, and 10 μM; P1884, TCI), cAMP inducers [Forskolin (10 μM FSK; F0855, TCI) and 3-Isobutyl-1-methylxanthine (10 μM IBMX; 095-03413, WAKO)], and cAMP analogs as PKA activator Sp-cAMP (10 μM; 14983, Cayman) and as EPAC/PKA activator 8-CPT-cAMP (10 μM; 12011, Cayman) to iHTMC. To analyze the antagonists for ARs, NE (5 μM) was simultaneously added with antagonists: β1β2-AR antagonist timolol maleate (1, 10, and 100 μM; T2905, TCI), β1-AR antagonist

betaxolol hydrochloride (1, 10, and 100 nM; B4474, TCI), β1-AR antagonist ICI-118.551 hydrochloride (0.01, 0.1, and 1 μM; HY-13951, MCE), and α2-AR antagonist phentolamine mesylate (100 μM; P1985, TCI). For pathway antagonists, dobutamine (1 μM) was simultaneously added with antagonists: PKA inhibitor KT5720 (0.001, 0.01, 0.1, and 1 μM; 10011011, Cayman), EAPC1/2 inhibitor ESI09 (0.1, 1, 10, and 100 μM; 19130, Cayman), PTEN inhibitor bpV (pic) (0.001, 0.01, 0.1, and 1 μM; SML0885, Sigma-Aldrich), and 3α-aminocholestane (0.01, 0.1, 1, and 10 μM; 3AC; HY-19776, MCE). To regulate PIP3 content, we treated iHTMC with the PIP3 antagonist PITenin-7 (0.01, 0.1, 1, and 10 μM; 524618, Calbiochem), PI3Kαβ inhibitor LY294002 (70920, Cayman), and selective PI3Kγ inhibitor CAY10505 (HY-13530, MCE).

**Western blot analysis**. Western blot analysis was performed as described previously in our report[86]. Protein extraction was performed using cell lysis buffer (9803, Cell Signaling Technology, Tokyo, Japan) containing a protease inhibitor cocktail (P8340, Sigma) and phosphatase inhibitor cocktail 1 (P2850, Sigma) according to the manufacturer's instructions. Total protein transferred to the PVDF membrane was detected using EzStainAQua MEM (WSE-7160, ATTO) and used for normalization. After destaining, membranes were incubated with the following primary antibodies: goat polyclonal antibody against ADRB1 (1:2000; NB600-978, Novus Biologicals), rabbit monoclonal antibodies against phospho-CREB (Ser133) (1:1000; 9198, Cell Signaling Technology), CREB (1:1000; 9192, Cell Signaling Technology), Akt (pan) (C67E7) (1:1000; 4691, Cell Signaling Technology), rabbit polyclonal antibody against phospho-PKCα/βII (Thr638/641) (1:1000; 9375, Cell Signaling Technology), phospho-Akt (Thr308) (1:1000; 9275, Cell Signaling Technology), phosphor-CaMKII alpha (Thr286) (1:1000; ab5683,abcam), ERK1/ERK2 (1:1000; A16686, ABclonal), phospho-ERK1(T202/Y204)/ERK2(T185/Y187) (1:1000; AP0472, ABclonal), INPP5D (SHIP1) (1:1000; A0122, ABclonal), phosphor-INPP5D (Tyr1021) (1:1000; PA903060, CSB), and mouse monoclonal antibody against CaMKIIα/β/γ/δ (G-1) (1:500; sc-5306, Santa Cruz Biotechnology). Membranes were washed and then incubated with HRP-conjugated goat polyclonal antibody against mouse and rabbit IgG (1:10,000; 7074, Cell Signaling Technology). Chemiluminescent images were detected using an Amersham Imager 600 (Cytiva Lifescience).

**cAMP measurement**. cAMP measurements were performed with a homogeneous TR-FRET immunoassay using the LANCE cAMP Detection Kit (AD0262, PerkinElmer, USA), according to the manufacturer's instructions (PerkinElmer). After confluence, iHTMCs in collagen I-coated 96-well microplates (4860-010; IWAKI) were washed with PBS (0.2 g/L EDTA), and then washed with stimulation buffer (HBSS, 5 mM HEPES, 0.5 mM IBMX, and 0.01% BSA at pH 7.4). After aspiration, iHTMC was added to 10 μL of tested compounds with FSK (0.001, 0.01, 0.1, and 1 μM), L-NE (0.01, 0.1, 1, and 10 μM), and dobutamine (0.001, 0.01, 0.1, 1, and 10 μM), along with 10 μL of Alexa Fluor 647 anti-cAMP antibody diluted with stimulation buffer. The cells were stimulated for 60 min at room temperature. The antagonist response analysis was performed using L-NE or dobutamine as the reference agonist. To analyze the antagonistic effect on β1-AR, the β1-AR agonist L-NE or dobutamine was used at submaximal concentrations (10 and 1 μM, respectively) to stimulate cAMP accumulation. These agonists and the β1-AR antagonist betaxolol (0.05, 0.5, 5, and 50 μM) were simultaneously added. After incubation, the reaction was stopped, and cells were lysed by the addition of 20 μL working solution (10 μL Eu-cAMP and 10 μL ULight-anti-cAMP), and incubated for 1 h at room temperature. The TR-FRET signal was read using a microplate reader SpectraMax M5 (Molecular Devices). cAMP concentrations were determined using GraphPad Prism software (version 6.0; GraphPad Software Inc., San Diego, CA, USA).

**Small interfering RNA knockdown**. iHTMCs were seeded in quadruplicate into collagen I-coated 96-well microplates (4860-010; IWAKI) at a density of $5.0 \times 10^3$ cells/well with serum-free TMCM (1% penicillin/streptomycin; #6591; Sciencell). Twenty-four hours later, cells were transfected with 0.04 μM Accell SMARTpool siRNA against *ADRB1, ADRB2, RAPGEF3, RAPGEF4, and PRKACA* (Supplementary Table 1, Dharmacon), Accell *GAPDH* pool-human siRNA (D-001930) was used as a positive control, and Accell Non-targeting pool siRNA (D-001910) as a negative control using 100 μL Accell Delivery Media (B-005000-100; Dharmacon), according to the manufacturer's instructions. Twenty-four hours later, NE (1 μM; 20 μL/well) or 0.1% DMSO with PBS containing pHrodo Green Zymosan Bioparticles (2.5 μg/well; P35365) were added. After 24 h of exposure to siRNA, total RNA was extracted and purified as described above for knockdown confirmation by qPCR.

**PIP3 extraction and quantification**. PIP3 measurements were performed using PIP3 Mass ELISA kit (K-2500s, Echelon Biosciences) according to the manufacturer's instructions. iHTMCs were seeded at a density of $1.2 \times 10^6$ cells/collagen I coated six well dish (IWAKI). After 14 h of treatment with dobutamine (1 μM) with or without agonists, KT5720 (10 μM), ESI09 (1 μM), bpV (pic) (1 μM), and 3AC (10 μM); the mediums were removed, and ice-cold 0.5 M tricarboxylic acid (1 mL) was immediately added. Scraped iHTMCs were transferred into a 1.5 mL tube, and centrifuged at 3000 rpm for 7 min at 4 °C. The pellet was resuspended in

5% TCA/1 mM EDTA (0.5 mL), vortexed for 30 s, and centrifuged at 3000 rpm for 5 min at room temperature. The washing step was repeated. Next, we added 0.5 mL of MeOH: CHCl3 (2:1) to extract neutral lipids, vortexed for 10 min at room temperature, and centrifuged at 3000 rpm for 5 min. We next added 0.5 mL MeOH: CHCl3: 12 M HCl (80:40:1) to extract the acidic lipids, vortexed for 25 min at room temperature, and centrifuged them at 3000 rpm for 5 min. After transferring the supernatant to a new 1.5 mL tube, 0.15 mL of CHCl3 and 0.27 mL of 0.1 M HCl were added, vortexed, and centrifuged at 3000 rpm for 5 min to separate organic and aqueous phases. The lower organic phase 0.3 mL was transferred into a new vial, and dried for 4 h at 4 °C. PIP3 samples were resuspended in 125 μL of PBS-Tween including 3% protein stabilizer (provided by the Echelon kit). Samples were sonicated in an ice-water bath for 5 min, vortexed, and spun down before being added to the ELISA. All experiments were performed three times, each performed in triplicate. The lipid amount (60 μL) was used and was run twice for each sample. After color reaction for 30 min in the dark, the 96-well plate was read by measuring the absorption at 450 nm with a SpectraMax M5 (Molecular Device). PIP3 concentrations were determined using GraphPad Prism software (version 6.0; GraphPad Software Inc., San Diego, CA, USA).

**Statistics and reproducibility**. Results are shown as the mean ± standard error of the mean (SEM) from at least three independent experiments and five mice. Statistical comparisons were made using GraphPad Prism 6 (GraphPad Software Inc., San Diego, CA) or Excel-Toukei 2012 software (Social Survey Research Information Co. Ltd., Osaka, Japan). Paired or Student's t-tests were used to compare two groups, and one-way analysis of variance (ANOVA) with Tukey's multiple comparison test or Dennett's multiple comparison test for more than three groups. Differences were considered statistically significant at $p < 0.05$.

**Reporting summary**. Further information on research design is available in the Nature Research Reporting Summary linked to this article.

## Data availability
Data of gene expression in human AH outflow-related cells have been deposited in the Gene Expression Omnibus accession number GSE146188[43]. All data is available from the corresponding author on reasonable request. Source values for each of the following figures are available in the corresponding Supplementary Data files: Fig. 1 - Supplementary Data 1; Fig. 2 - Supplementary Data 2; Fig. 3 - Supplementary Data 3; Supplementary Fig. 4 - Supplementary Data 4; Fig. 5 - Supplementary Data 5; Fig. 6 - Supplementary Data 6; Fig. 7 - Supplementary Data 7; Supplementary Fig. 1 - Supplementary Data 8; Supplementary Fig. 2 - Supplementary Data 9; Supplementary Fig. 3 - Supplementary Data 10; Supplementary Fig. 5 - Supplementary Data 11; Supplementary Fig. 6 - Supplementary Data 12.

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

## Acknowledgements

We are grateful to the Institute of Comprehensive Medical Research, Division of Animal Research Promotion Division (Aichi Medical University), for maintaining the mice, providing advanced research promotion for the expression analysis of genes and proteins, and performing measurements of cAMP, fluorescence, and phagocytosis. We also thank Dr. Sachiko Takamura (Department of Microbiology and Immunology, Aichi Medical University) for their valuable advice. This study was supported by the JSPS KAKENHI (Grant Number 19K09962), the 24th General Assembly of the Japanese Association of Medical Sciences, Terumo Life Science Foundation, Takeda Science Foundation, Suzuken Memorial Foundation, Kato Memorial Bioscience Foundation, the UBE Foundation, and Toyoaki Scholarship Foundation.

## Author contributions

K.I. contributed conceptualization, methodology, investigation, writing original draft, funding acquisition. K.I. and S.M. contributed writing review & editing, resources, supervision.

## Competing interests

The authors declare no competing interests.
