## [Peer Review File · Communications Biology]

Reviewers' comments:

Reviewer #1 (Remarks to the Author):

Brief summary of the manuscript

This manuscript identifies and describes the intracellular signaling pathway involved in regulation of trabecular meshwork phagocytosis under the treatment of sympathetic norepinephrine to understand the molecular basis for the nocturnal increase in intraocular pressure. Increased nocturnal intraocular pressure (IOP) in humans and diurnal animals of normal and glaucoma subjects is well recognized however; the molecular basis for nocturnal rise in IOP is poorly understood. Using the transformed cell line of human trabecular meshwork and live mouse models, the authors, in this study, have methodically and mechanistically investigated, norepinephrine suppressed phagocytosis in the trabecular meshwork cells, which are involved in aqueous humor drainage and homeostasis of intraocular pressure. This study characterizes, identifies and reports the role for β 1-aderenergic receptor mediated cAMP-EPAC-SHIP1 activation in norepinephrine suppressed phagocytosis and nocturnal rise in IOP by inhibiting PIP3-dependent phagocytosis.

Overall impression of the work

Overall, this is an important study addressing the physiological regulation of nocturnal rise in IOP, which is expected to have a potential significance for developing the treatments for increased nocturnal IOP in glaucoma patients. The authors have conducted several cell-based experiments in conjunction with rodent (mouse) animal model to understand how sympathetic norepinephrine and adrenal glucocorticoids regulated trabecular meshwork phagocytosis influence aqueous humor outflow and nocturnal change in IOP. Although, the reported results support the involvement of norepinephrine activated β 1-aderenergic receptor mediated cAMP-EPAC-SHIP1 signaling pathway in decreased trabecular meshwork cell phagocytosis activity, and agonists of this pathway increasing nocturnal IOP, there are several concerns for the described conclusions. Additional studies and revision are required to strengthen the author's conclusions.

Specific comments, with recommendations for addressing each comment

1. The authors need to confirm the validity of trabecular meshwork cell line used in the described study. This is not a well-characterized and used cell line and not sure, whether this line retains the some of the well- recognized characteristics of human trabecular meshwork cells. Especially, since the authors did not find the previously reported findings of the glucocorticoids effects on phagocytosis. At minimum, the authors need to conform whether the cell line used in this study induces expression of myocilin, a glucocorticoid induced protein, which is preferentially expressed in the trabecular meshwork within the aqueous humor drainage pathway. Glucocorticoids are known to influence the actin cytoskeletal organization and cell adhesion in various cell lines including in trabecular meshwork cells, and these changes are well recognized to impact phagocytosis in various cell types. However, in this study, the authors could not confirm this well-recognized activity of glucocorticoids in the trabecular meshwork cells.

2. The proposed β 1-aderenergic receptor mediated cAMP-EPAC-SHIP1 signaling pathway and PIP3 in phagocytosis are expected to have a significant influence on the actin cytoskeletal organization and cell adhesion, which are known to have direct effect on the aqueous humor outflow through the trabecular pathway. Therefore, it is uncertain whether the author's interpretation that suppressed phagocytosis resulted from the activation of β 1-aderenergic receptor- mediated cAMP-EPAC-SHIP1 signaling pathway is the cause for nocturnal IOP rise. For example, in this study, the authors have used Cytochalasin D to suppress phagocytosis. Cytochalasin D has been shown to lower IOP. Similarly, Rho kinase inhibitor, which depolymerizes actin cytoskeleton, is used to treat ocular hypertension in glaucoma patients. Therefore, it is necessary to test the effects of various agonists and antagonists used in this study on the trabecular meshwork cells and distinguish the cytoskeletal and phagocytosis effects on aqueous humor outflow and IOP.

3. In support of the proposed role for the β 1-aderenergic receptor mediated cAMP-EPAC-SHIP1 signaling pathway in nocturnal IOP rise, there should be an adequate explanation from the authors regarding why the double knockout mouse model of β 1 and β 2 adrenergic receptors showed no effect on nocturnal IOP published by other investigators.

4. The authors have used several agonists and antagonists topically on mice and report significant

changes on IOP. There is not much published data on the effects of these agents on mouse models. The authors need to cite prior references in support of their findings to confirm the responsiveness of the used compounds on IOP in a mouse model.

5. The authors need to use a well-established agent that is known to suppress phagocytosis as a positive agent to use topically and confirm its effects on IOP (both nocturnal and daytime).

Reviewer #2 (Remarks to the Author):

In the paper entitled "Involvement of trabecular meshwork phagocytic suppression by sympathetic norepinephrine in nocturnal intraocular pressure rise", the authors investigate the role of NE and dexamethasone in regulating the phagocytic properties of an immortalized TMC cell line.

This study has merit and provides stimulating findings for further research activities on the topic, but there are some issues that need to be addressed to improve the quality of the manuscript and the rigor through which the experiments have been designed.

- Why ouabain was injected in the posterior chamber of the eye and not in the anterior chamber?

- Is there any specific reason why in experiment using beads the zeitgeber time was fixed at ZT6 and ZT15 instead of ZT10 and ZT15 as previously done for IOP determination in the presence of ouabain?

- Figure 1H is not commented in the text.

- Line 143: the authors missed out to specify that Dexa was delivered in combination with NE, I suppose.

- For what concerns the results shown in Figure 2, there is just one point that looks suffering from a poor conceptual consistency. NE half-life is about 2 minutes in vivo. I acknowledge that NE half life in cell culture medium may be longer than in serum, but how can a single NE dose account for a driven output rhythm over several hours? In Figure 2 caption (line 885) the authors refer to a "continuous NE treatments", does it mean that fresh NE was delivered several times during the time course? If so, this is not specified in the results and methods sections.

- Dexamethasone stimulation introduces many metabolic effects on TMCs, often resulting in cell senescence, especially after a prolonged stimulation of cells. Data regarding Dexa should be interpreted with caution.

- For what concerns the RNA-interference experimental procedures, there is a major issue that must be addressed. The authors verify the effectiveness of selected genes silencing by monitoring the mRNA at 24 h from antisense delivery by RT-PCR. This is ok, but it does not provide a good readout about effective protein downregulation, that is what really matters. In most cases, siRNA experiments are run over 72-96 h for this reason. I do see that to address this point may require additional time and costs, I am not asking to repeat the whole experimental procedure of such a complex work, but effective downregulation of proteins (at least a 50% reduction with respect to untreated cells) should be verified.

- Line 202: "insufficiently but significantly rescued..." this is not clear. If there is a statistically significant difference, this should be clarified in figures and throughout the text.

- For what concerns PKA and EPAC1/2 antagonists, the time of stimulation of the cells with the two drugs must be indicated.

- Total protein staining in the case of Western blotting is inadequate. The lanes showing the full pattern of proteins used for protein normalization should be provided at least in suppl. Info.
- The whole paper, especially introduction and discussion, should be shortened out.

Response to editor's and reviewers' comments

Response to editor's and reviewers' comments to Ikegami et al., "Involvement of trabecular meshwork phagocytic suppression by sympathetic norepinephrine in nocturnal intraocular pressure rise" (COMMSBIO-21-3193-T)

We are grateful for the editor's and reviewers' constructive comments, which helped us to significantly improve our manuscript.

Below, the editor's and reviewers' comments are shown in black text, and our answers to the comments are shown in blue text.

Editor:

Please address all comments raised by the reviewers as much as in feasibly possible. In particular please ensure you fully respond to reviewer 1's concerns regarding cell line validation and the proposed β 1-aderenergic receptor mediated pathway as well as reviewer 2's point regarding the level of downregulation. If feasible, we also required additional experiments using a well-established agent that is known to suppress phagocytosis (reviewer 1, point 5).

Response: Thank you for providing the advices. We have revised the manuscript as indicated by both reviewers. For the cell validation proposed by Reviewer 1, we verified the induction of Myocilin gene expression after Dex stimulation. In addition, as Reviewer 1 suggests, we have examined the effect of established phagocytotic inhibitor (dynasore) on increasing IOP during the day and at night. By pharmacological examination of the effect of IOP regulation on the cytoskeleton in a time-dependent manner, we clarified the involvement of phagocytosis in the increase in AH outflow during the daytime. Furthermore, following the comments of Reviewer 2, we verified the suppression of the protein by siRNA using only ADRB1, because of cost.

Reviewers' comments:

Reviewer #1 (Remarks to the Author):

Brief summary of the manuscript

This manuscript identifies and describes the intracellular signaling pathway involved in regulation of trabecular meshwork phagocytosis under the treatment of sympathetic norepinephrine to understand the molecular basis for the nocturnal increase in

intraocular pressure. Increased nocturnal intraocular pressure (IOP) in humans and diurnal animals of normal and glaucoma subjects is well recognized however; the molecular basis for nocturnal rise in IOP is poorly understood. Using the transformed cell line of human trabecular meshwork and live mouse models, the authors, in this study, have methodically and mechanistically investigated, norepinephrine suppressed phagocytosis in the trabecular meshwork cells, which are involved in aqueous humor drainage and homeostasis of intraocular pressure. This study characterizes, identifies and reports the role for β 1-aderenergic receptor mediated cAMP-EPAC-SHIP1 activation in norepinephrine suppressed phagocytosis and nocturnal rise in IOP by inhibiting PIP3-dependent phagocytosis.

Overall impression of the work

Overall, this is an important study addressing the physiological regulation of nocturnal rise in IOP, which is expected to have a potential significance for developing the treatments for increased nocturnal IOP in glaucoma patients. The authors have conducted several cell-based experiments in conjunction with rodent (mouse) animal model to understand how sympathetic norepinephrine and adrenal glucocorticoids regulated trabecular meshwork phagocytosis influence aqueous humor outflow and nocturnal change in IOP. Although, the reported results support the involvement of norepinephrine activated β 1-aderenergic receptor mediated cAMP-EPAC-SHIP1 signaling pathway in decreased trabecular meshwork cell phagocytosis activity, and agonists of this pathway increasing nocturnal IOP, there are several concerns for the described conclusions. Additional studies and revision are required to strengthen the author's conclusions.

Response: We would like to thank reviewer 1 for their insightful comments and suggestions. For the cell validation proposed by Reviewer 1, we verified the induction of Myocilin gene expression after Dex stimulation. In addition, as Reviewer 1 suggests, we examined the effect of established phagocytotic inhibitors on increasing IOP during the day and at night. We also examined whether the IOP induction by β 1-AR agonist dobutamine at day is canceled by the actin polymerization inhibitors and phagocytosis activator to distinguish the cytoskeletal and phagocytosis effects on AH outflow and IOP. As reviewer 1 suggested, additional studies and revision made the author's conclusions strengthen.

Specific comments, with recommendations for addressing each comment

1. The authors need to confirm the validity of trabecular meshwork cell line used in the described study. This is not a well-characterized and used cell line and not sure, whether this line retains the some of the well- recognized characteristics of human trabecular meshwork cells. Especially, since the authors did not find the previously reported findings of the glucocorticoids effects on phagocytosis. At minimum, the authors need to conform whether the cell line used in this study induces expression of myocilin, a glucocorticoid induced protein, which is preferentially expressed in the trabecular meshwork within the aqueous humor drainage pathway. Glucocorticoids are known to influence the actin cytoskeletal organization and cell adhesion in various cell lines including in trabecular meshwork cells, and these changes are well recognized to impact phagocytosis in various cell types. However, in this study, the authors could not confirm this well-recognized activity of glucocorticoids in the trabecular meshwork cells.

Response: Thank you for your constructive comments. As you suggested, we validated the induction of *MYOC* expression by Dex in this iHTMC (Supplementary Figure 1). We also analyzed the expression of *PER1*, which is known to directly be induced by GC, and *MMP3*, which is a marker of TM cells known to suppress expression by Dex. As a result, *PER1* was induced early, and *MYOC* expression was drastically and significantly increased by 48 hours exposure of Dex, while *MMP3* was suppressed. This result means that this iHTMC retains the properties of TMC. We have changed as follows:

In the Results

“We first validated these iHTMCs by analyzing gene expression (direct GC-induced clock gene *PER1*, trabecular meshwork–inducible GC response protein [*MYOC*]³⁶, and GC-suppressed matrix metalloproteinase *MMP3*^{37,38}), 6 h (early) and 48 h (late) after Dex stimulation. Early induction of *PER1*, late induction of *MYOC*, and late suppression of *MMP3* were detected (Supplementary Fig. 1), indicating that this iHTMC has TM cell properties³⁹.” (p 7, line 150-155)

↑Supplementary Figure 1 (we added a new data as this figure)

2. The proposed β 1-aderenergic receptor mediated cAMP-EPAC-SHIP1signaling pathway and PIP3 in phagocytosis are expected to have a significant influence on the actin cytoskeletal organization and cell adhesion, which are known to have direct effect on the aqueous humor outflow through the trabecular pathway. Therefore, it is uncertain whether the author's interpretation that suppressed phagocytosis resulted from the activation of β 1-aderenergic receptor- mediated cAMP-EPAC-SHIP1signaling pathway is the cause for nocturnal IOP rise. For example, in this study, the authors have used Cytochalasin D to suppress phagocytosis. Cytochalasin D has been shown to lower IOP. Similarly, Rho kinase inhibitor, which depolymerizes actin cytoskeleton, is used to treat ocular hypertension in glaucoma patients. Therefore, it is necessary to test the effects of various agonists and antagonists used in this study on the trabecular meshwork cells and distinguish the cytoskeletal and phagocytosis effects on aqueous humor outflow and IOP.

Response: Thank you for your advice. We agree that this is an important point.

We examined the effect of established phagocytotic inhibitors on increasing IOP during the day and at night. We also examined whether the IOP induction by β 1-AR agonist dobutamine at day is canceled by the actin polymerization inhibitors and phagocytosis activator to distinguish the cytoskeletal and phagocytosis effects on AH outflow and IOP. Since changes in the cytoskeleton sometimes interacts with phagocytosis, we cannot completely separate it from phagocytosis. Therefore, to think separately as your advice, we analyzed the effect of dynamin inhibitor dynasore as the established phagocytosis

inhibitor, cytochalasin D / latrunculin A, which suppress the phagocytosis by inhibiting cytoskeleton polymerization, and RKI1447, which suppresses cytoskeletal polymerization and activate phagocytosis by inhibiting ROCK, on the IOP in the light and dark periods (Fig. 2; below). As a result, dynasore increased IOP only in the light period, suggesting the involvement of phagocytosis in daytime-increased AH outflow, consistent with the result in Fig.1.

↑Figure 2 (we added a new data as Fig. 2)

↑Figure 7a-f (we modified Fig.7 d-f)

In contrast, nocturnal IOP rise were suppressed by the inhibitors except dynasore (Fig. 2). In addition, when we gave these drugs to mice before dobutamine, only ROCK inhibitor suppressed dobutamine-induced IOP rise in the light period (Fig. 7; above). Taken together, these results suggest that changes in AH outflow by cytoskeletal dynamics seems to be limited to nighttime. We concluded that the time-dependent AH outflow regulations by TM phagocytosis can be distinguished from the mechanism by cytoskeletal dynamics to some extent by day and night. Therefore, the results and discussions were added as follows.

Introduction

“Conversely, actin remodeling, such as fragmentation or polymerization, also regulates AH outflow, and the actin cytoskeleton of TM cells is a therapeutic target in glaucoma patients. Small GTPase Rho-associated coiled-coil-containing protein kinase (ROCK) inhibitors lower IOP, not only by relaxation of the TM by disruption of actin stress fibers, but also by the activation of phagocytosis^{19–21}. However, since all phagocytic processes are driven by a finely controlled rearrangement of the actin cytoskeleton^{22,23}, it has been difficult to verify the effects of these two factors separately.” (p 4-5, line 81-88)

“...and can promote actin polymerization in retina²⁷.” (p 5, line 93)

In the Results

“TM phagocytosis is involved in diurnal IOP reduction

Phagocytosis and remodeling of the actin cytoskeleton are related to AH outflow^{15,19-21}. Dynasore, an inhibitor of dynamin GTPase activity, but not other small GTPases, has been widely studied in clathrin-mediated endocytosis and phagocytosis, also in HTMC³¹ (Fig. 2a). Dynamin is thought to interact with actin filaments when the edges of the phagocytic cup close³². Latrunculin A and cytochalasin D, drugs that interfere with actin–myosin contraction, also lower IOP^{33,34} and suppress TM phagocytosis³³⁻³⁵ (Fig. 2a). To examine their involvement in daytime IOP reduction, we first administered mice dynasore, cytochalasin D, latrunculin A, and RKI1447 (ROCK inhibitor; phagocytosis enhancer and cytoskeleton disruptor [Fig. 2a]) at ZT4 and measured IOP at ZT9 (Fig. 2b). Instillation with dynasore significantly increased IOP, whereas latrunculin A decreased IOP (Fig. 2c). Individual data also showed that dynasore enhanced diurnal IOP (Fig. 2d), indicating that the daytime activation of TM phagocytosis may suppress IOP, which can be canceled by depolymerization of actin fibers. In contrast, instillation with cytochalasin D, latrunculin A, and RKI1447 at ZT10 arrested the nocturnal IOP increase, especially latrunculin A significantly suppressed IOP, but not dynasore (Fig. 2e, f). At the individual level, these drugs, regardless of their effects on phagocytosis, also suppressed the nocturnal IOP rise (Fig. 2g), indicating the role of TM actin polymerization in nocturnal IOP rise. This IOP suppression limited to nighttime by ROCK inhibitor was similar to a previous report demonstrating that RhoA blocking by AAV prevents nocturnal IOP elevation in rats³⁶. Taken together, these results suggest that AH outflow regulation by the cytoskeleton seems to be night-limited or time-independent, while TM phagocytosis may mediate the daytime increase in AH outflow.” (p 6-7, line 124-146)

“To clarify the involvement of cytoskeletal rearrangement and phagocytosis in this enhancement, we next evaluated cytochalasin D, latrunculin A, and RKI1447. Interestingly, only RKI1447, a phagocytosis promoter, prevented this increase (Fig. 7e,f), indicating the contribution of TM phagocytosis in β 1-AR-mediated IOP increase rather than actin rearrangement. In other words, the decrease in AH outflow by phagocytosis suppression outweighed that of

actin rearrangement in the other TM cells. Conversely, since these inhibitors suppressed nocturnal IOP increase (Fig. 2), the increase in AH outflow by actin depolymerization may increase at night.” (p 14, line 316-318)

In the Discussion

“In the present study, the time-dependent mechanism of the inhibitory effect of NE on TM phagocytosis was elucidated, but the circadian regulation of NE/GC in other important key factors such as cytoskeleton, cell adhesion, and IOP-independent uveoscleral outflow to the ciliary muscle (Fig. 6i) in the AH outflow remain unknown. Long-term stimulation of GC increases the actin polymerization of TM and induces fibrosis⁶⁶. In the TM, cAMP/PKA activation and downstream RhoA inactivation lead to a loss of actin stress fibers and focal adhesions and disassembly of the matrix network⁶⁷. *Ship1* *-/-* in neutrophils upregulates basal actin polymerization⁶⁸. However, the time-dependent efficacy may explain their contribution to some extent. The effects of actin polymerization inhibitors on IOP reduction seem to be limited to the dark period in mice (Fig 2; Fig 7). In fact, the decrease in AH outflow by phagocytosis suppression on day appears to outweigh that by actin rearrangement in the TM. The cell-intrinsic circadian clock regulates numerous cytoskeletal regulators in fibroblast⁶⁹. In contrast, in mice, 30–42% of AH passes through the uveoscleral pathway⁷⁰. To understand the AH dynamic rhythm completely, these determinants need to be elucidated. Discovery of a phagocytosis activator independent of the cytoskeleton or single cell analysis separated by the function of TM will be able to elucidate them completely in the future. ” (p 16, line 350-365)

In addition, to validate phagocytosis assay *in vitro*, we used Dynasore instead of cytochalasin D. We added the results as follows:

In the Results

“Next, the validity of this phagocytosis assay was confirmed using a control with Dynasore (Fig. 3b).” (p 7, line 157-158)

↑Figure 3 (we modified Fig. 3b from cytochalasin D to well-established phagocytosis inhibitor Dynasore)

3. In support of the proposed role for the β 1-aderenergic receptor mediated cAMP-EPAC-SHIP1signaling pathway in nocturnal IOP rise, there should be an adequate explanation from the authors regarding why the double knockout mouse model of β 1 and β 2 adrenergic receptors showed no effect on nocturnal IOP published by other investigators.

Response: Since, in previous study using the double knockout mouse model of β 1 and β 2 ARs, this mice had different genetic background compared with the control mice, we think it impossible to assess whether the IOP rhythm was attenuated or not. However, it is certain that β 1- and β 2-ARs are not essential for IOP rhythm formation. In present study, Betaxolol did not prevent nocturnal IOP completely (Fig. 7), so the contribution of β 1-AR-mediated nocturnal AH resistance for IOP rhythm formation may be not so high, and the other circadian factors may regulate AH-outflow/inflow. We corrected the discussion as follows.

In the Discussion

“Since, in a previous study demonstrating that the double knockout mouse model of $\beta 1$ and $\beta 2$ ARs maintains IOP rhythm ⁷, these mice had different genetic backgrounds compared with the control mice, we think it is impossible to assess the IOP rhythmicity such as its amplitude. However, it is certain that $\beta 1/2$ -ARs are not essential for IOP rhythm formation ⁷. In the present study, betaxolol did not completely prevent nocturnal IOP rise (Fig. 7). The removal of SCG in mice attenuated the IOP rhythm ¹³, and betaxolol instillation resulted in IOP rhythms with nocturnal peak flattening in both POAG and NTG patients ⁵⁷. The contribution of the $\beta 1$ -AR-mediated circadian rhythm of AH resistance for IOP rhythm formation may not be so high, and Dex or the other AR can regulate AH-outflow/inflow. $\beta 2$ -AR- or GC-mediated AH production in the NPE of the ciliary body is related to the IOP rhythm ¹³. This may explain why β -AR1/2 double knockout mice maintain this IOP rhythm.” (p 17, line 375-385)

4. The authors have used several agonists and antagonists topically on mice and report significant changes on IOP. There is not much published data on the effects of these agents on mouse models. The authors need to cite prior references in support of their findings to confirm the responsiveness of the used compounds on IOP in a mouse model.

Response: Since we could not find published data on the effects of several agents on mouse IOP, topical administration of reagents in present study may be a first demonstration. However, it needs to be elucidated whether these effects on IOP changes act on TM phagocytosis or not in future. We added the explanation and instead added references about the effect of the agents on phagocytosis or eye as follows:

Dynasore (clathrin-mediated endocytosis and phagocytosis inhibitor):

In the Results

“Dynasore, an inhibitor of dynamin GTPase activity, but not other small GTPases, has been widely studied in clathrin-mediated endocytosis and phagocytosis, also in HTMC ³¹ (Fig. 2a). Dynamin is thought to interact with actin filaments when the edges of the phagocytic cup close ³².” (p 6, line 126-129)

Latrunculin A and cytochalasin D

In the Results

“Latrunculin A and cytochalasin D, drugs that interfere with actin–myosin contraction, also lower IOP^{33,34} and suppress TM phagocytosis^{33–35} (Fig. 2a).” (p 6, line 129-130)

Betaxolol β 1-AR antagonist:

In the Results

“First, the β 1-AR antagonist betaxolol is well used for glaucoma therapy⁵⁷, lowers topically mouse IOP⁵⁸, it blocked the nocturnal IOP increase (Fig. 7a).” (p 13, line 290-291)

ESI09 (EPAC1 inhibitor) and KT5720 (PKA inhibitor):

In the Results

“Furthermore, blocking of PKA and EPAC1/2 by antagonists (KT5720 and ESI09, respectively) prevents β 2-AR- and PGE2-suppressed neutrophil phagocytosis⁴⁷” (p 13, line 313-314)

“Furthermore, the effects of KT5720, ESI09, and bpV(pic) on retinal photoreceptor death in rodents^{59,60} have been reported but not on IOP, while that of the 3AC, promoting phagocytosis⁵⁴, on IOP in animals remains unknown.” (p 13, line 293-295)

bpV(pic) (PTEN inhibitor), 3AC and K118 (SHIP1 inhibitor):

In the Results

“Furthermore, the effects of KT5720, ESI09, and bpV(pic) on retinal photoreceptor death in rodents^{59,60} have been reported but not on IOP, while that of the 3AC, promoting phagocytosis⁵⁴, on IOP in animals remains unknown.” (p 13, line 293-295)

5. The authors need to use a well-established agent that is known to suppress phagocytosis as a positive agent to use topically and confirm its effects on IOP (both nocturnal and daytime).

Response: Thank you for your grateful advice. As we describe above (Reviewer 1 No.2),

we have used a well-established phagocytosis inhibitor that is Dymanin inhibitor Dynasore to confirm the diurnal and nocturnal effects on IOP by eye drop. We added the results as follows:

In the Results

“TM phagocytosis is involved in diurnal IOP reduction

Phagocytosis and remodeling of the actin cytoskeleton are related to AH outflow^{15,19-21}. Dynasore, an inhibitor of dynamin GTPase activity, but not other small GTPases, has been widely studied in clathrin-mediated endocytosis and phagocytosis, also in HTMC³¹ (Fig. 2a). Dynamin is thought to interact with actin filaments when the edges of the phagocytic cup close³². Latrunculin A and cytochalasin D, drugs that interfere with actin–myosin contraction, also lower IOP^{33,34} and suppress TM phagocytosis³³⁻³⁵ (Fig. 2a). To examine their involvement in daytime IOP reduction, we first administered mice dynasore, cytochalasin D, latrunculin A, and RKI1447 (ROCK inhibitor; phagocytosis enhancer and cytoskeleton disruptor [Fig. 2a]) at ZT4 and measured IOP at ZT9 (Fig. 2b). Instillation with dynasore significantly increased IOP, whereas latrunculin A decreased IOP (Fig. 2c). Individual data also showed that dynasore enhanced diurnal IOP (Fig. 2d), indicating that the daytime activation of TM phagocytosis may suppress IOP, which can be canceled by depolymerization of actin fibers. In contrast, instillation with cytochalasin D, latrunculin A, and RKI1447 at ZT10 arrested the nocturnal IOP increase, especially latrunculin A significantly suppressed IOP, but not dynasore (Fig. 2e, f). At the individual level, these drugs, regardless of their effects on phagocytosis, also suppressed the nocturnal IOP rise (Fig. 2g), indicating the role of TM actin polymerization in nocturnal IOP rise. This IOP suppression limited to nighttime by ROCK inhibitor was similar to a previous report demonstrating that RhoA blocking by AAV prevents nocturnal IOP elevation in rats³⁶. Taken together, these results suggest that AH outflow regulation by the cytoskeleton seems to be night-limited or time-independent, while TM phagocytosis may mediate the daytime increase in AH outflow.” (p 6-7, line 124-146)

Reviewer #2 (Remarks to the Author):

In the paper entitled "Involvement of trabecular meshwork phagocytic suppression by

sympathetic norepinephrine in nocturnal intraocular pressure rise", the authors investigate the role of NE and dexamethasone in regulating the phagocytic properties of an immortalized TMC cell line.

This study has merit and provides stimulating findings for further research activities on the topic, but there are some issues that need to be addressed to improve the quality of the manuscript and the rigor through which the experiments have been designed.

- Why ouabain was injected in the posterior chamber of the eye and not in the anterior chamber?

Response: Thank you for your pointing out. This is our typographical error. In the method, it was described as the anterior chamber, but in the text it was the posterior chamber. We changed them and have modified the methods as follow:

In the Results

"...were treated with an intraocular injection of an Na⁺/K⁺ATPase inhibitor ouabain (100 μM/0.1% dimethyl sulfoxide [DMSO], phosphate-buffered saline [PBS], 3 μL) into anterior chamber of the right eye with a 34-gauge needle (0.18 × 8 mm, Pasny; Unisis) connected to a Hamilton syringe at ZT10. To precisely control the small volume (3 μL) of anterior chamber injection, 3 μL PBS (0.1% DMSO) was injected into the left eye with a 34-gauge needle connected to a Hamilton syringe." (p 20, line 452-458)

- Is there any specific reason why in experiment using beads the zeitgeber time was fixed at ZT6 and ZT15 instead of ZT10 and ZT15 as previously done for IOP determination in the presence of ouabain?

Response: Thank you for your comment. In previous several studies, the peak of IOP in mice is about ZT15 and the trough is ZT6. IOP is still low on the ZT10. Therefore, in the bead experiment, the IOP peak ZT15 and the trough ZT6 are used for measurement. On the other hand, in an experiment to verify the effect of the drugs on the nocturnal IOP increase, problems such as intraocular clearance occur when time is left after administration, so we calculated back from the IOP peak ZT15, and decided to administer it to ZT10 just before the dark period when IOP is low and time is not left until

nocturnal IOP measurement. This can also be said for eye drop experiments in Figure 6. We added sentences in Methods as follows:

Methods

“For the analysis of phagocytosis-related drugs and dobutamine-mediated IOP induction, IOP was measured before drug instillation at zeitgeber time (ZT) 4 and measured at ZT9. ZT0 (0800) was defined as the time of light ON. To analyze the nocturnal IOP increase, IOP was obtained by measuring the IOP at ZT10 and ZT15. We calculated back from ZT15 (the peak of IOP) for 5 h and set it to ZT10 when IOP was low. For diurnal changes in IOP, IOP was measured at ZT6 (IOP trough) and ZT15 (IOP peak),¹³ 2 weeks after bead injection.” (p 19-20, line 441-447)

- Figure 1H is not commented in the text.

Response: Thank you. We added sentences in Results as follows:

In the Results

“..., where we assumed that the particles were taken into the SC or passed through the TM (Fig. 1h). The fluorescence intensity at ZT6 was significantly stronger than that at ZT18 (Fig. 1h,1i), consistent with a previous rabbit AH outflow study³⁰.” (p 6, line 117-119)

- Line 143: the authors missed out to specify that Dexa was delivered in combination with NE, I suppose.

Response: Thank you for your suggestion. Since we think this interested and important mechanism for glaucoma therapy, we will sophisticate this mechanism in future. We changed sentences in Results and Discussion as follows:

In the Results

“Dex with NE-suppressed phagocytosis in a dose-dependent manner (Supplementary Fig. 3b), consistent with previous *in situ* studies^{24,25}. Long-term Dex exposure did not alter cell viability (Supplementary Fig. 3c) or senescence (Supplementary Fig. 3d), indicating no effect of apoptosis or senescence on

phagocytosis inhibition by Dex with NE. GC also binds to TM in humans ⁴¹, and its receptor localizes in mouse TM ¹³. Thus, the interaction between GCs and NE can generate an appropriate AH drainage rhythm.” (p 7, line 163-171)

In the Discussion

“In the present study, we found that NE was a necessary condition for GC-mediated phagocytosis in the TM. Both β -adrenergic signaling and GCs are mediators of SCN timing signals in osteoblasts ⁷⁵. Interactions between the sympathetic nervous system and GCs have also been previously reported. In particular, GC transcriptionally modulates β 2-AR expression by modulating GC-response elements on the promoter ⁷⁶. Interestingly, GCs rapidly activate cAMP production via G α s to initiate non-genomic signaling, which contributes to one-third of their canonical genomic effects ⁷⁷. In fact, betaxolol prevented steroid-induced IOP increase ⁷⁸. Thus, in TM, G α s-bound GR may enhance the β 1-AR-G α s signal to suppress phagocytosis and generate an appropriate AH drainage rhythm. ” (p 17, line 386-394)

- For what concerns the results shown in Figure 2, there is just one point that looks suffering from a poor conceptual consistency. NE half-life is about 2 minutes in vivo. I acknowledge that NE half life in cell culture medium may be longer than in serum, but how can a single NE dose account for a driven output rhythm over several hours?

Response: Thank you for your comment. As you concern about half-life of NE in medium, we measured it in iHTMC cultured medium. As a result, we found far longer half-life of NE (~18 h) than in vivo. This result indicates that *in vitro* NE treatment continuously act on cells. Since NE is released with a nocturnal peak from the SCG in rodents, nocturnal NE elevation rhythm can regulate diurnal changes of TM phagocytosis. We changed arrows to bar in Figure 7d (below), and added the explanation, data (Supplementary Fig. 2; below) and the results as follows:

↑Figure 7d-f (we modified Fig. 7d)

↑Supplementary Figure 2 (we added new data as this figure)

In the Results

“When we performed an *in vitro* NE clearance assay, the half-life of medium NE (~18.6 h, $y = 335.9 e^{-0.000621 x}$, $r^2 = 0.979$) (Supplementary Fig. 2) was far greater than that of blood NE (a few minutes), suggesting that *in vitro* NE treatment continuously act on cells.” (p 8, line 159-162)

“Circadian time signals can generate rhythmicity through self-sustainable autonomous rhythm by single stimulation or a driven-output system by stimulation of circadian factors such as NE and GCs once a day (Fig. 3g). Since NE from the SCG appears to show a nocturnal peak in rodents⁴², the circadian rhythm of SCG-NE can regulate diurnal changes in TM phagocytosis.” (p 8, line 172-176)

“Since NE from the SCG appears to show a nocturnal peak in rodents⁴², the circadian rhythm of SCG-NE can regulate diurnal changes in TM phagocytosis (Fig. 7d). (Fig. 7d).” (p 14, line 311-313)

In Figure 2 caption (line 885) the authors refer to a “continuous NE treatments”, does it mean that fresh NE was delivered several times during the time course? If so, this is not specified in the results and methods sections.

Response: In present study, “continuous NE treatments” means without washout of medium NE and without medium changes. We added and changed sentences in Methods as follows:

In the Methods

“..., and were exposed over 3 days without washout.” (p 23, line 534)

Figure legends

“Continuous NE exposure dose-dependently prevented phagocytic activity in iHTMC during 2 days of culture” (p 40, line 976-977)

- Dexamethasone stimulation introduces many metabolic effects on TMCs, often resulting in cell senescence, especially after a prolonged stimulation of cells. Data regarding Dexa should be interpreted with caution.

Response: Thank you for your comment. We agree that this is an important point. When we measured Dex-induced cell viability and senescence marker beta-galactosidase level in iHTMC, we could not detect significant changes in these (Supplementary Fig. 3; below). These results indicate that Dex stimulation did not result in cell senescence in this study. We changed sentences in Results and Discussion as follows:

In the Results

“Long-term Dex exposure did not alter cell viability (Supplementary Fig. 3c) or senescence (Supplementary Fig. 3d), indicating no effect of apoptosis or senescence on phagocytosis inhibition by Dex with NE.” (p 8, line 166-168)

↑Supplementary Figure 3 (we added c,d about cell viability and senescence)

- For what concerns the RNA-interference experimental procedures, there is a major issue that must be addressed. The authors verify the effectiveness of selected genes silencing by monitoring the mRNA at 24 h from antisense delivery by RT-PCR. This is ok, but it does not provide a good readout about effective protein downregulation, that is what really matters. In most cases, siRNA experiments are run over 72-96 h for this reason. I do see that to address this point may require additional time and costs, I am not asking to repeat the whole experimental procedure of such a complex work, but effective downregulation of proteins (at least a 50% reduction with respect to untreated cells) should be verified.

Response: Thank you for your comment. We measured protein downregulation of ADRB1 by siRNA using western blotting. As a result, we verified significant down regulation of ADRB1 (~48 %) at 24 h of siRNA exposure (Supplementary Figure 6; below). We added data and sentences in Results as follows:

In the Results

“*ADRB1* siRNA significantly suppressed *ADRB1* protein levels (Supplementary Fig. 6b)” (p 10, line 207-208)

↑Supplementary Figure 6 (we added b,c about *ADRB1* down-regulation by RNAi)

- Line 202: “insufficiently but significantly rescued...” this is not clear. If there is a statistically significant difference, this should be clarified in figures and throughout the text.

Response: Thank you for your comment. In the statistical analysis, the L-NE significantly suppressed phagocytosis as compared with the control group, and there was no significant difference between the L-NE group and the siRNA group. However, since siRNA restored phagocytosis to a control level ($p > 0.05$ vs control group), so we corrected it as follows:

In the Results

“RNA interference against *PRKACA*, *RAPGEF3*, and *RAPGEF4*, encoding PKA, EPAC1, and EPAC2, respectively, rescued β 1-AR-suppressed iHTMC phagocytosis to a control level ($p > 0.05$, vs. control; Fig. 5i).” (p 10, line 232-234)

- For what concerns PKA and EPAC1/2 antagonists, the time of stimulation of the cells with the two drugs must be indicated.

Response: Thank you for your comment. These antagonists were simultaneously added with dobutamine without preincubation to prevent endogenous circadian clock in the iHTMC from being reset or affected by these reagents. We added the information about

the time of stimulation of PKA and EPAC1/2 antagonists in the result, method, and figure legends.

In the Methods

“For pathway antagonists, dobutamine (1 μ M) was simultaneously added with antagonists” (p 24, line 553-554)

In Figure legends

“The recovery effect of PKA and EPAC1/2 antagonists (KT5720 and ESI09, respectively) simultaneously added with dobutamine (1 μ M) dose-dependently rescued dobutamine-suppressed iHTMC phagocytosis” (p 42, line 1036-1039)

- Total protein staining in the case of Western blotting is inadequate. The lanes showing the full pattern of proteins used for protein normalization should be provided at least in suppl. Info.

Response: According nature guideline, we added full pattern of Western blotting in Supplementary Figure 7 (below).

Fig. 5b Western blots used for densitometry quantification

Fig. 6e Western blots used for densitometry quantification

Fig. 6f Western blots used for densitometry quantification

↑Supplementary Figure 7

- The whole paper, especially introduction and discussion, should be shortened out.

Response: Thank you. We shortened Introduction and Discussion.

REVIEWERS' COMMENTS:

Reviewer #1 (Remarks to the Author):

The authors have adequately addressed all the concerns raised by the two reviewers by including the additional data derived from the new experiments. This revised manuscript is in much better state than the initial version in supporting the conclusions of the study.

Reviewer #2 (Remarks to the Author):

The manuscript has been significantly improved and in this version is suitable for publication.

Author responses

Reviewer #1 (Remarks to the Author):

The authors have adequately addressed all the concerns raised by the two reviewers by including the additional data derived from the new experiments. This revised manuscript is in much better state than the initial version in supporting the conclusions of the study.

>Thank you for your kind and constructive reviewing.

Reviewer #2 (Remarks to the Author):

The manuscript has been significantly improved and in this version is suitable for publication.

> Thanks for the thought-provoking peer review.